# Learning State Representations via Retracing in Reinforcement Learning

**Changmin Yu**[1]*, **Dong Li**[2], **Jianye Hao**[3, 2], **Jun Wang**[1, 2], **Neil Burgess**[1]*

[1]UCL, London, United Kingdom
[2]Huawei Noah's Ark Lab
[3]College of Intelligence and Computing, Tianjin University
{changmin.yu.19; n.burgess}@ucl.ac.uk;
{lidong106; w.j}@huawei.com; jianye.hao@tju.edu.cn

## Abstract

We propose *learning via retracing*, a novel self-supervised approach for learning the state representation (and the associated dynamics model) for reinforcement learning tasks. In addition to the predictive (reconstruction) supervision in the forward direction, we propose to include "retraced" transitions for representation/model learning, by enforcing the cycle-consistency constraint between the original and retraced states, hence improve upon the sample efficiency of learning. Moreover, *learning via retracing* explicitly propagates information about future transitions backward for inferring previous states, thus facilitates stronger representation learning for the downstream reinforcement learning tasks. We introduce *Cycle-Consistency World Model* (*CCWM*), a concrete model-based instantiation of *learning via retracing*. Additionally we propose a novel adaptive "truncation" mechanism for counteracting the negative impacts brought by "irreversible" transitions such that *learning via retracing* can be maximally effective. Through extensive empirical studies on visual-based continuous control benchmarks, we demonstrate that *CCWM* achieves state-of-the-art performance in terms of sample efficiency and asymptotic performance, whilst exhibiting behaviours that are indicative of stronger representation learning.

## 1 Introduction

Recent developments in deep reinforcement learning (RL) have made great progress in solving complex control tasks (Mnih et al., 2013; Levine et al., 2016; Silver et al., 2017; Vinyals et al., 2017; Schrittwieser et al., 2020). With the increasing capacity of deep RL algorithms, the problems of interests become increasingly complex. An immediate challenge is that the observation space becomes unprecedentedly high-dimensional, and often the perceived observations have significant redundancy and might only contain partial information with respect to the associated ground-truth states, hence negatively impacting the policy learning. The field of representation learning offers a wide range of approaches for extracting useful information from high-dimensional data (with potentially sequential dependency structure) (Bengio et al., 2013). Many recent works have explored the application of representation learning in RL (Ha and Schmidhuber, 2018; Hafner et al., 2019; 2020a; Schrittwieser et al., 2020; Schwarzer et al., 2021; Zhang et al., 2020), which lead to superior performance comparing to naive embedding. Many such algorithms rely on predictive (reconstruction) supervision for representation learning, such that the effects of actions in the observable space are maximally preserved in the learned representation space.

Here we argue that existing methods do not fully exploit the supervisory signals inherent in the data. Additional valid supervision can often be obtained for representation learning by including temporally "backward" transitions in situations in which the same set of rules govern both temporally forward and backward transitions. Hence, with "*learning via retracing*", we obtain more training samples for representation learning without additional interaction with the environment (twice as much as existing approaches in tasks that admit perfect reversibility across all transitions). Therefore, we hypothesise that by augmenting representation learning with "learning via retracing", we can significantly improve

---

*Please send any enquiry to changmin.yu.19@ucl.ac.uk and n.burgess@ucl.ac.uk

the sample efficiency of representation learning and the overall RL task, which is a long-standing issue that plagues the practical applicability of deep RL algorithms. Beyond improved sample efficiency, joint predictive supervision in temporally forward and backward directions use information from both the future and the past for the inference of states, similar to the smoothing operation for latent state inference in state-space models (Kalman, 1960; Murphy, 2012), leading to more accurate latent state inference, hence achieving stronger representation learning.

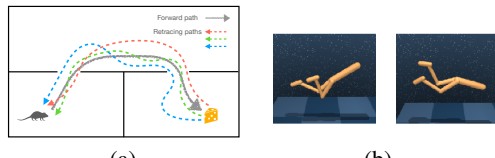

(a)                                          (b)

Figure 1: **Motivation of "learning via retracing".** (a): Retracing in navigation tasks yields faster representation learning and potentially supports stronger generalisation; (b): "Irreversible" transitions (graphical demonstration from the DeepMind Control Suite Tassa et al. (2018)).

As a motivating example, consider a rat navigating towards a cheese in a cluttered environment (Figure 1a). Upon first visit to the goal state, multiple imaginative retracing trajectories can be randomly simulated. By constraining the *temporal cycle-consistency* of the retracing transitions, the rat quickly builds a state representation that accurately preserves the transitions in the area around the actual forward trajectory taken by the rat. Moreover, all retracing simulations pass through the two "doors", allowing the rat to quickly identify the key bottleneck states that are essential for successful navigation towards the goal and generalisation to other task topologies (Section 5.3). We conjecture that such imaginative retracing could be neurally implemented by the reversed hippocampal "replay" that has been observed in both rodents and humans (Foster and Wilson, 2006; Penny et al., 2013; Liu et al., 2021) (see Section 6 for further discussion).

One problem that hinders the successful application of "learning via retracing" is that reversibility might not be preserved across all valid transitions, i.e., there exists transitions such that $s \rightarrow s'$ for some action $a$, but no action $a'$ such that $s' \rightarrow s$ (Figure 1b). Under these situations, naively enforcing the similarity between the representations of $s$ and $\check{s}$ (the retraced state given $s'$ and $a$, see Section 3) leads to a suboptimal representation space, potentially hindering RL training. Hence in order to maximally preserve the advantages brought by "learning via retracing", it is essential to identify "irreversible" transitions and rule them out from representation learning via retracing. To this end, we propose a novel dynamically regulated approach for identifying such "irreversible" states, which we term adaptive truncation (Section 3.3).

"Learning via retracing" can be integrated into any representation learning method that utilises a transition model, under both the model-free and model-based RL frameworks. Here we propose *Cycle-Consistency World Model* (CCWM), a self-supervised instantiation of "learning via retracing", for joint representation learning and generative model learning under the model-based RL setting. We empirically evaluate CCWM on challenging visual-based continuous control benchmarks. Experimental results show that CCWM achieves state-of-the-art performance in terms of sample efficiency and asymptotic performance, whilst providing additional benefits such as stronger generalisability and extended planning horizon, indicative of stronger representation learning is achieved.

## 2 Preliminaries

### 2.1 Problem Formulation

We consider reinforcement learning problems in Markov Decision Processes (MDPs). An MDP can be charaterised by the tuple, $\mathcal{M} = \langle \mathcal{S}, \mathcal{A}, \mathcal{R}, \mathcal{P}, \gamma \rangle$, where $\mathcal{S}, \mathcal{A}$ are the state and action spaces, respectively; $\mathcal{R} : \mathcal{S} \rightarrow \mathbb{R}$ is the reward function (we assume deterministic reward functions unless stated otherwise), $\mathcal{P} : \mathcal{S} \times \mathcal{A} \times \mathcal{S} \rightarrow [0,1]$ is the transition distribution of the task dynamics; $\gamma \in \mathbb{R}$ is the discounting factor. The control policy, $\pi : \mathcal{S} \times \mathcal{A} \rightarrow [0,1]$, represents a distribution over actions at each state. The goal is to learn the optimal policy, $\pi^*$, such that the expected future reward is maximised across all states, i.e.,

$$\pi^* = \arg\max_{\pi \in \Pi} \mathbb{E}_\pi \left[ \sum_t \gamma^t \mathcal{R}(s_t, a_t) | s_0 = s \right], \forall s \in \mathcal{S}, \text{ where } s_t \sim \mathcal{P}(\cdot | s_{t-1}, a_{t-1}) \text{ for } t = 1, 2, \ldots$$

(1)

Here we consider tasks in which the perceivable observation space, $\mathcal{O}$, is high-dimensional, due to either redundant information or simply because only visual inputs are available. Hence it is necessary

to learn an embedding function, $\phi : \mathcal{O} \rightarrow \mathcal{Z}$, such that the embedded observation space, $\phi\mathcal{O}$, could act as $\mathcal{S}$ in the MDP, to support efficient learning of the optimal policy using existing RL algorithms.

## 2.2 GENERATIVE MODELLING OF DYNAMICS

Modelling the transition dynamics using a sequential VAE-like structure enables joint learning of the latent representation and the associated latent transitions. Specifically, the dynamics model is defined in terms of the following components (see also top part in Figure 2b).

$$\text{Observation (context) embedding: } e_t = q_\phi(O_t),$$
$$\text{Latent posterior distribution: } p(z_{t+1}|z_t, a_t, O_{t+1}),$$
$$\text{Latent variational transition (prior) distribution: } q_{\psi_1}(z_{t+1}|z_t, a_t), \quad (2)$$
$$\text{Latent variational posterior distribution: } q_{\psi_2}(z_{t+1}|z_t, a_t, e_{t+1}),$$
$$\text{Generative distribution: } p_\theta(O_{t+1}|z_{t+1}),$$

where $\psi = \{\psi_1, \psi_2, \phi\}$ and $\theta$ represent the parameters associated with the recognition and generative models respectively. Latent variables $z \in \mathcal{Z}$ are introduced for more flexible modelling of the distribution of the observed variables. A variational approximation is employed since the true posterior $p(z_{t+1}|z_t, a_t, O_{t+1})$ is usually intractable in practice.

At each time step $t$, the agent receives an observation $O_t$, which is then embedded into a context vector $e_t = q_\phi(O_t)$. After initialisation, the latent space vector is rolled out in a forward fashion given the action $a_t$, yielding one-step prediction into the future following the variational transition (prior) distribution $q_{\psi_1}(z_{t+1}|z_t, a_t)$ (we assume standard first-order Markovian structure in the latent space). The dynamics model is trained via maximum likelihood learning, and due to intractability, we adopt standard amortised variational inference, and the parameters of the recognition and generative models in Eq. 2 are learned by maximising the variational free energy (also known as the ELBO; (Wainwright and Jordan, 2008; Higgins et al., 2016)).

$$\mathcal{L}_{\text{forward}}(O_{t+1}) = \mathbb{E}_{z_{t+1}\sim q_{\psi_2}}[\log p_\theta(O_{t+1}|z_{t+1}) - \beta\mathcal{D}_{\text{KL}}\left[q_{\psi_2}(z_{t+1}|z_t, a_t, e_{t+1})||q_{\psi_1}(z_{t+1}|z_t, a_t)\right]], \quad (3)$$

The variational free energy objective consists of the reconstruction error, $p_\theta(O_{t+1}|z_{t+1})$, and the KL-divergence between the variational posterior distributions and the predictive prior as regularisation. The intuitive autoencoder-like structure nicely separates the generative process from the inference process, with the variational posterior ($q_{\psi_2}(z_{t+1}|z_t, a_t, e_{t+1})$) serving as the main inference engine of the optimal latent representations. Note that the $\beta$ parameter controls the degree of factored structure (disentanglement) of the latent code, which by default is set to 1 (Higgins et al., 2016).

## 3 METHOD

We firstly introduce *learning via retracing* in its most general format, then provide a concrete model-based instantiation based on generative dynamics modelling We finally propose a novel adaptive truncation scheme for dealing with the "irreversibility" issue in "learning via retracing".

### 3.1 LEARNING VIA RETRACING

We always assume the usage of an approximate dynamics model, regardless of the overall RL agent being model-free or model-based (the dynamics model would only be used for representation learning under the model-free setting, hence it is possible to have separate dynamics models for forward and "reversed" transitions). Given a set of observations, $\mathbf{O} = \{O_1, \ldots, O_T\}$, a dynamics model, $\mathcal{M} : \mathcal{S} \times \mathcal{A} \times \mathcal{S} \rightarrow [0, 1]$, most existing methods of self-supervised representation learning involve learning an encoder ($\mathcal{E}_\phi$) and a decoder ($\mathcal{D}_\theta$), trained via minimising the predictive reconstruction.

$$\mathcal{L}(\phi, \theta) = d(O_{1:T}, \mathcal{D}_\theta(f(\mathcal{E}_\phi(O_{1:T}); A_{1:T-1}, \mathcal{M})) \quad (4)$$

where $d$ is some metric of the observable space (e.g., the $L2$ distance). The function $f(e; \mathcal{M}, a)$ is some function specifying the schedule for predictions, e.g., $f(e_t, \mathcal{M}, a_t) = \mathcal{M}(e_t, a_t) = \hat{e}_{t+1}$ corresponds to learning the representations based on one-step predictive reconstruction.

Existing methods explore predictive supervision in a temporally forward direction, however, we argue that the "reversed" transitions can also contain useful signals for learning. Consider a transition tuple, $(s, a, s')$, we define the "reversed" transitions being the tuple $(s', a', s)$, where $a'$ is the "reversed" action. In situations where the same set of rules apply to forward and backward transitions, the reversed transition given $a'$ could contribute to representation learning via Eq. 4 as an additional training

sample (utilising a potentially different loss function from the forward supervision, see Figure 2a).

Hence by utilising the additional reversed transition for representation learning, we improve the sample efficiency of learning without additional interaction with the environment. As mentioned previously, in situations where perfect reversibility is not preserved across all transitions, such "irreversible" transitions could negatively impact the overall learning. Correct identification of such states is hence essential for the successful implementation of "learning via retracing". To this end, we propose a novel adaptive truncation scheme in Section 3.3.

Despite the intuitive simplicity of *learning via retracing*, it offers a number of advantages comparing to existing representation learning methods. In addition to the improved sample efficiency, we can interpret learning with "reversed" transitions as explicit inference of the current latent state given future information. In combination with the forward predictive supervision, the joint learning dynamics is similar to the smoothing operation in dynamical systems, which is often superior than filtering (corresponds to using solely the forward predictive supervision) in terms of inference accuracy (Kalman, 1960; Murphy, 2012). Hence *learning via retracing* could support stronger representation learning for the downstream RL task. We note that the

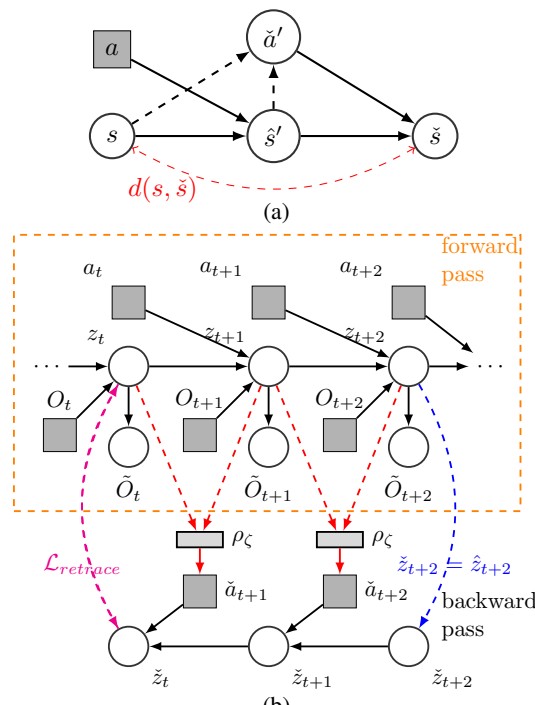

Figure 2: **Graphical illustration of "learning via retracing".** (a) "learning via retracing" additionally constrains the similarity between the retraced and original states for representation learning; (b) Graphical model of *CCWM*. The empty circle and filler square nodes represent the stochastic and deterministic variables, respectively.

overall RL agent could still benefit from the representations obtain from learning via retracing even in tasks without perfect reversibility, such as a moving car where the external state feature, such as the absolute location, and controllable internal state features, such as the velocity and acceleration, are not jointly reversible. In such cases, we expect learning via retracing would dissect out the controllable internal features from the external features and prioritise the training with respect to such features.

We have introduced *learning via retracing* in its most general form, where a large degree of freedom exists such that the method can be tailored and integrated with many of the existing representation learning approaches. There are many free model choices, such as being model-free or model-based; whether or not to use a separate "reversed" dynamics model; loss function for constraining the "retraced" transitions; deterministic or probabilistic dynamics model, just to name a few. Below we provide one concrete instantiation of *learning via retracing*, the *Cycle-Consistency World Model* (*CCWM*), under the model-based framework based on generative dynamics modelling.

## 3.2 CYCLE-CONSISTENCY WORLD MODEL

*CCWM* is a model-based RL agent that utilises a generative world-model, trained given both the predictive reconstruction of future states, and constraining the temporal cycle-consistency of the "retraced" states (i.e., constraining the retraced states to match the original states). For notational convenience, we denote all predictive prior estimates, posterior estimates, and the retraced predictive latent estimates as $\hat{z}$, $\tilde{z}$ and $\check{z}$, respectively.

We use the similar dynamics model described in Section 2.2 (top panel in Figure 2b). We additionally define a reverse action approximator, $\rho_\zeta : \mathcal{Z} \times \mathcal{Z} \to \mathcal{A}$, which takes in a tuple of latent states $(z_{t+1}, z_t)$ and outputs an action $\check{a}_{t+1}$ that approximates the "reversed" action that leads the transition from $z_{t+1}$ back to $z_t$. Instead of introducing a separate "reversed" dynamics model, we use the same dynamics model for both the forward and retracing transitions, which lead to improved sample efficiency of model learning in addition to representation learning. The parameters of $\rho$, $\zeta$, can be

either learned jointly with the model in an end-to-end fashion, or trained separately (see Appendix D). The graphical model of *CCWM* is shown in Figure 2b.

During training, given a sample trajectory $\{O_{1:T+1}, a_{1:T}\}$, we firstly perform a one-step forward sweep through all timesteps to compute the variational prior and posterior estimates of latent states.

$$
\begin{aligned}
\hat{z}_{\tau+1} &\sim q_{\psi_1}(z|\hat{z}_\tau, a_\tau) \\
\tilde{z}_{\tau+1} &\sim q_{\psi_2}(z|\hat{z}_\tau, a_\tau, q_\phi(O_\tau))
\end{aligned}
\tag{5}
$$

for $\tau = 0, \ldots, T$, where $\hat{z}_0$ is randomly initialised. Note that we also include reward prediction as part of the dynamics modelling, and we have omitted showing this for simplicity.

Given the predictive estimates in the forward direction, we compute the "retracing" estimates, utilising the same latent transition dynamics (variational predictive prior distribution).

$$
\check{z}_\tau \sim q_{\psi_1}(z|\tilde{z}_{\tau+1}, \check{a}_{\tau+1}), \text{ where } \check{a}_{\tau+1} = \rho_\zeta(\tilde{z}_{\tau+1}, \tilde{z}_\tau), \text{ for } \tau = 1, \ldots, T
\tag{6}
$$

The forward and retracing predictive supervision separately contributes to the model learning of *CCWM*. For the forward pass, the parameters of the dynamics model are trained to maximise the likelihood of the sampled observations via predictive reconstruction. We follow the variational principle, by maximising the variational free energy (Eq. 3), computed by Monte Carlo estimate given the posterior predictive samples. For the "retracing" operation, model learning is based on constraining the deviation of the retraced states from the original states. Intuitively, this utilises the temporal cycle-consistency of the transition dynamics: assuming that the action-dependent transition mapping is invertible across all timesteps (Dwibedi et al., 2019). The loss function for constraining the cycle-consistency is another degree of freedom of "learning via retracing". For *CCWM*, we choose *bisimulation metric* as the loss function for the retracing operations, which has been shown to yield stronger constraints of the latent states on the MDP level, and also leads to more robust and noise-invariant representation without reconstruction (Ferns et al., 2011; Zhang et al., 2020).

$$
\mathcal{L}_{\text{retrace}}(\tilde{z}_t, \check{z}_t) = \mathbb{E}_{\check{z}_t}\left[(||\tilde{z}_t - \check{z}_t||_1 - \mathcal{D}_{\text{KL}}[\hat{R}(\cdot|\tilde{z}_t)||\hat{R}(\cdot|\check{z}_t)] - \gamma W_2(q_{\psi_1}(\cdot|\tilde{z}_t, \pi(\tilde{z}_t)), q_{\psi_1}(\cdot|\check{z}_t, \pi(\check{z}_t))))^2\right],
\tag{7}
$$

where $\hat{R}(r|z)$ represents the learned reward distributions (we assume stochastic rewards), and $W_2(\cdot, \cdot)$ represents the 2-Wasserstein distance (see Eq. 11 in Appendix D). The advantage of choosing the bisimulation metric as the retrace loss function is further empirically shown in Appendix G through careful ablation studies (Figure 11). Multiple retracing trajectories can be simulated and the retrace loss is again a Monte Carlo estimate based on sampled "retracing" states, but empirically we observe that *one* "retracing" sample is sufficient as we do not observe noticeable improvements for increasing the number of "retraced" samples. We note that here we utilise the same transition dynamics model for both forward and reversed rollouts, which might cause issues in model learning due to the absence of perfect "reversibility" across all valid transitions. Hence we need a method for dynamically assessing the "reversibility" so as to know when to apply learning via retracing (see Section 3.3).

The overall objective for the *CCWM* dynamics model training is thus a linear combination of the forward and "retracing" loss functions.

$$
\mathcal{L}(\theta, \psi, \zeta) = \frac{1}{NT} \sum_{n=1}^{N} \sum_{\tau=1}^{T} \left[\mathcal{L}_{\text{forward}}(O_\tau^n; \theta, \psi) + \lambda \mathcal{L}_{\text{retrace}}(\tilde{z}_\tau^n, \check{z}_\tau^n; \psi, \zeta)\right],
\tag{8}
$$

where $\lambda$ is the scalar multiplier for the retrace loss, and $N$ is the batch size. We implement *CCWM* using a Recurrent State-Space Model (RSSM; Hafner et al. (2019)). The complete pseudocode for *CCWM* training is shown in Algorithm 1 in Appendix A. We note that "learning via retracing" is also applicable under the model-free setting, we describe one such instantiation in Appendix B.

## 3.3 REVERSIBILITY AND TRUNCATION

As we noted above, perfect reversibility is not always present across all transitions in many environments. For instance, consider the falling android presented in Figure 1b, it is trivial to observe that no valid action is able to transit a falling android to its previous state. Under such situations, naive application of "learning via retracing", by constraining the temporal cycle-consistency, will corrupt

representation learning (and dynamics model learning in *CCWM*). Here we propose an approach to deal with such "irreversibility".

Our approach is based on adaptive identification of "irreversible" transitions. We propose that the value function of the controller (e.g., an actor-critic agent) possesses some information about the continuity of the latent states (Gelada et al. (2019); see Appendix C for further discussion). Hence we use the value function as an indicator for sudden change in the agent's state. Specifically, for each sampled trajectory, we firstly compute the values of each state-action pair using the current value function approximator, $[Q(z_1, a_1), \ldots, Q(z_T, a_T)]$. We then compute the averages of the values over a sliding window of size $S$ through the value vectors of each sampled trajectory, resulting in a $(T - S)$-length vector $[\bar{Q}_1, \ldots, \bar{Q}_{T-S}]$. Any drop/increase in the sliding averages (above some pre-defined threshold) indicates a sudden change in the value function, hence a sudden change in the latent representation given the continuity conveyed by the value function. Given some timestep, $\tau$, at which the sudden change occurs, we then remove the transitions $\{z_{\tau-S:\tau}, a_{\tau-S:\tau}\}$ from "learning via retracing". Such adaptive scheduling allows us to deal with "irreversibility".

## 4 RELATED WORKS

**Representation learning in RL.** Finding useful state representations that could aid RL tasks has long been studied. Early works have investigated representations based on a fixed basis such as tile coding and Fourier basis (Mahadevan, 2005; Sutton and Barto, 2018). With the development of deep learning techniques, recent works explored automatic feature discovery based on neural network training, which can be categorised into three large classes. The first class of methods studies the usage of data augmentation for broadening the data distribution for training more robust feature representation (Laskin et al., 2020; Kostrikov et al., 2020; Schwarzer et al., 2021; Yarats et al., 2021). The second class explores the role of auxiliary tasks in learning representations, such as weakly-supervised classification and location recognition, for dealing with sparse and delayed supervision (Lee et al., 2020b; Mirowski et al., 2017; Oord et al., 2018). The third class of methods, specifically tailored to model-based RL models, leverages generative modelling of environment dynamics, enabling joint learning of the representations and the dynamics model (Ha and Schmidhuber, 2018; Buesing et al., 2018; Hafner et al., 2019; 2020a; Lee et al., 2020a; Schrittwieser et al., 2020; Hafner et al., 2020b).

**Cycle-Consistency.** Cycle-consistency is a commonly adopted approach in computer vision and natural language processing (Zhou et al., 2016; Zhu et al., 2017; Yang et al., 2017; Dwibedi et al., 2019), where the core idea is the validation of matches between cycling through multiple samples. We adopt similar design principles for sequential decision-making tasks: rollouts in a temporally forward direction alone yield under-constrained learning of the world model. By additionally incorporating backwards rollouts into model learning in a self-supervised fashion, we enforce the inductive bias that the same transition rules govern the dynamics of the task.

Concurrent to our work, Yu et al. (2021) proposed *PlayVirtual*, a model-free RL method that integrates a similar cycle-consistency philosophy into training representations with data augmentations (Schwarzer et al., 2021). We note that *PlayVirtual* falls under the proposed "learning via retracing" framework, but lying on the opposite spectrum comparing to the *CCWM* agent, being model-free and utilising a separate reversed dynamics model, with the latter being the main difference between the premises of PlayVirtual and CCWM. By using the same dynamics model for both the forward and reversed transitions, we hope to exploit and embed the context prior of reversible transitions into the learnt representations, hence the induced advantages extend beyond the explicit advantage of improved sample efficiency, but also stronger generalisation, improved predictive rollouts (leads to more accurate policy gradient hence improving policy training).

## 5 EXPERIMENTAL STUDIES

The experimental studies aim at examining if "learning via retracing" truly helps with the overall RL training and planning, improves the generalisability of the learned representation, and whether the truncation schedule proposed in Section 3.3 deals with the irreversibility of some state transitions.

### 5.1 EXPERIMENT SETUP

*CCWM* can be combined with any value-based or policy-based RL algorithms. We implement *CCWM* with a simple actor-critic RL agent with generalised advantage estimation based on standard model-

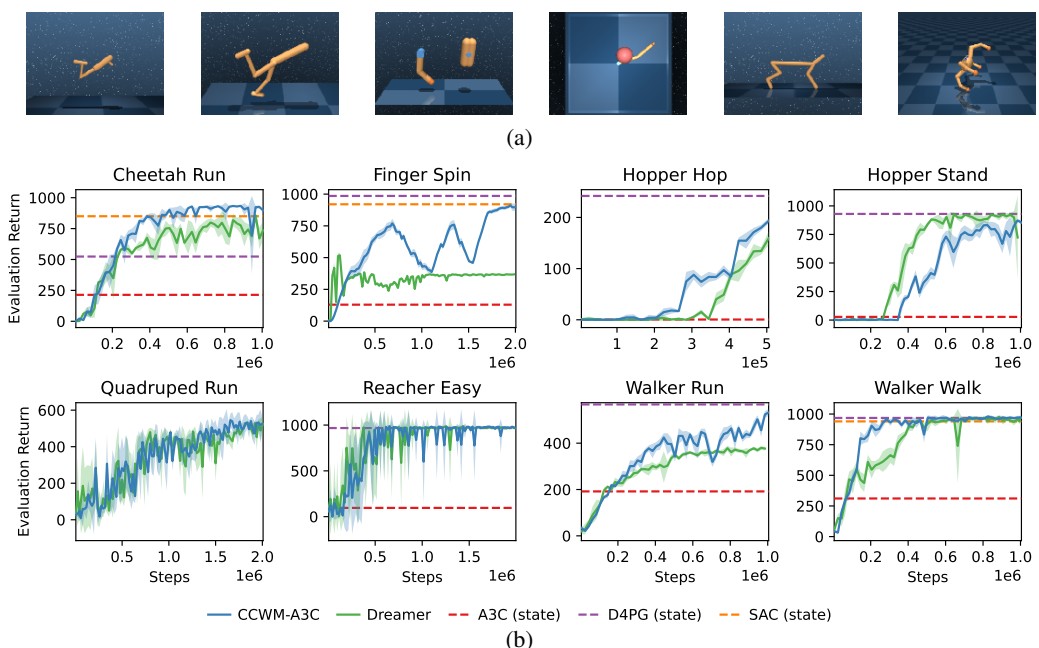

Figure 3: **Evaluation of *CCWM* on DeepMind Control Suite.** (a): Graphical demonstration of selected continuous control task environments, from left to right: hopper stand/hop, walker run/walk, finger spin, reacher easy, cheetah run, quadruped run. (b): Average evaluation returns ($\pm 1$ s.d.) during training (5 random seeds). "Learning via retracing" generally improves the performance of learning from pixel inputs in presented tasks comparing to the main baseline Dreamer agent (which could approximately be viewed as *CCWM* without retracing). *CCWM* reaches the asymptotic performance of state-of-the-art model-free methods (SAC, D4PG at $10^8$ steps) on several tasks.

based RL framework using model-based rollouts[1] (Sutton and Barto, 2018; Konda and Tsitsiklis, 1999; Schulman et al., 2015). We base our experimental studies on the challenging visual-based continuous control benchmarks for which we choose 8 tasks from the DeepMind Control Suite (Tassa et al. (2018); Figure. 3a). The details of training and the architecture can be found in Appendix D.

**Baselines**: We implement Dreamer as our main model-based baseline (Hafner et al., 2020a), which represents the current state-of-the-art world-model-type model-based RL agent on visual-based continuous control tasks. We also compare with the following model-free baselines: SAC Haarnoja et al. (2018), D4PG Barth-Maron et al. (2018), A3C Mnih et al. (2016). We implement the SAC agent given the state inputs and directly report the asymptotic performance of the D4PG and A3C algorithms from Tassa et al. (2018). We report the asymptotic scores for the model-free algorithms due to the large gap in sample efficiency comparing to the model-based methods.

### 5.2 Evaluation on Continuous Control Tasks

The performance of *CCWM* and selected baseline algorithms is shown in Figure 3b. The empirical results show that *CCWM* (without adaptive truncation introduced in Section 3.3) generally achieves faster behaviour learning comparing to the baselines, which conforms with our hypothesis that utilising backward passes in addition to forward passes provides additional supervision, hence improving the sample efficiency of learning (see also Appendix E). *CCWM* outperforms Dreamer on 5 of the selected tasks, and is comparable to Dreamer on 2 of the remaining 3 tasks, in terms of both the sample efficiency and final convergence performance. We note the relative poor performance of CCWM on the Hopper Stand task, which might be accredited to the inherent large degree of irreversibility of the task, we shall examine how to deal with such irreversible tasks in Section 5.4.

Moreover, in the "Cheetah Run" task, *CCWM* converges at $\sim 900$ score with $\sim 5 \times 10^5$ steps, whereas Dreamer, by the time it has received $1 \times 10^6$ training steps, is yet to reach a comparable

---

[1]Note that for the maximally fair comparison with our main baseline, Dreamer (Hafner et al., 2020a), we utilise the same policy agent as Dreamer in order to fully demonstrate the utility of "learning via retracing" .

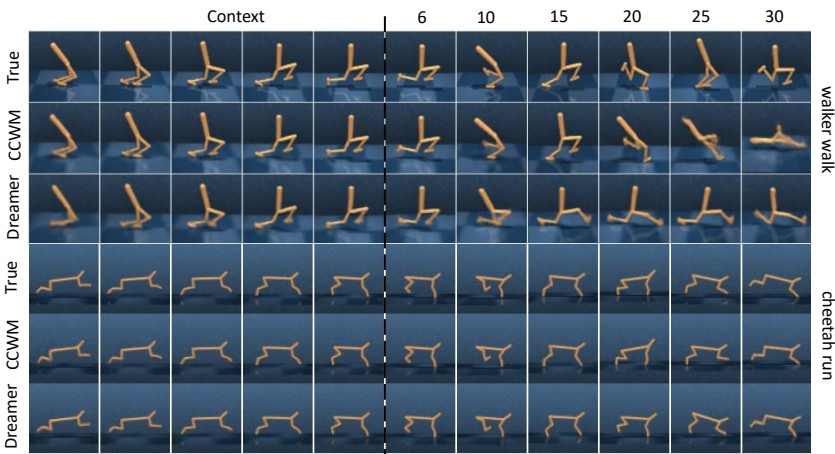

Figure 4: **Qualitative comparison of long-range predictive reconstruction of *CCWM* and Dreamer.** Predictive rollouts over 30 time-steps given the actions are computed using the representation models. *CCWM* consistently generates more accurate predictive reconstructions further into the future than Dreamer, with *CCWM* becoming noticeably inaccurate by $25 - 30$ timesteps, and Dreamer by $10 - 15$ timesteps. See implementation details and further discussion in Appendix F.

| CHANGED COMPONENTS | CCWM (MEAN $\pm 1$ S.D.) | DREAMER (MEAN $\pm 1$ S.D.) | P-VALUE (3 S.F.) | SIGNIFICANT? ($\alpha = 0.01$) |
|---|---|---|---|---|
| R + M + F | $\mathbf{628.07 \pm 36.95}$ | $468.82 \pm 94.73$ | $\mathbf{7.27 \times 10^{-6}}$ | YES |
| R + M + S + F | $\mathbf{641.89 \pm 28.67}$ | $562.58 \pm 93.91$ | $\mathbf{3.97 \times 10^{-3}}$ | YES |

Table 1: Evaluation of trained *CCWM* and Dreamer on the ability of zero-shot transfer in cheetah run tasks with different configurations. (R: Reward; M: Mass; F: Friction; S: Stiffness.)

score. This demonstrates that "learning via retracing" brings more benefits beyond plain sample efficiency, i.e., doubling the training steps does not eliminate the performance gap. This corresponds to our hypothesis that by explicitly conveying future information back to previous states, "learning via retracing" enables the learning of task-aware representations that support stronger behaviour learning. To test our hypothesis, we empirically evaluate *CCWM*'s ability of long-term predictive reconstructions and compare with Dreamer. To ensure fair comparison, we provide further training for Dreamer whenever necessary, such that the asymptotic performance is comparable with *CCWM* (see Appendix D for details). Figure 4 shows that *CCWM* consistently yields more accurate predictive reconstructions over a longer time span, on both the walker walk and cheetah run tasks. The empirical evidence confirms our hypothesis that by incorporating "learning via retracing" into model learning enables the resulting latent space to support more accurate latent predictions, hence leading to stronger behaviour learning. Increased range of accurate latent prediction additionally enables *CCWM* to perform better planning. We provide further analysis of predictive reconstruction in Appendix F.

### 5.3 ZERO-SHOT TRANSFER

Based on the motivation that "learning via retracing" will improve the generalisability of the agent (Figure 1a), we empirically test the generalisability of *CCWM* on the basis of zero-shot transfer tasks. Specifically, we modify a number of basic configurations of the cheetah run task, such as the mass of the agent and the friction coefficients between the joints. The details of the changes can be found in Appendix D. Despite the increased sample efficiency of *CCWM* over Dreamer during training (Figure 3b), both methods converge at similar values at $2 \times 10^6$ steps. We directly evaluate the trained agents on the updated cheetah run task without further training to test their abilities on zero-shot transfer. We report the mean evaluation scores ($\pm$ 1 s.d.) of both agents over 15 random seeds, as well as the one-sided t-test statistics and significance of the difference between the two sets of evaluations in Table 1. The overall performance on zero-shot transfer of our approach is comparable with Dreamer on simpler transfer tasks (full results shown in Appendix 5.3), and significantly outperforms Dreamer on more non-trivial modifications to the original task. The introduction of retracing also improves the stability of zero-shot transfer in general (reduced variance in evaluation). These confirm our previous hypothesis that "learning via retracing" improves the ability of within-domain generalisation.

## 5.4 ADAPTIVE TRUNCATION OF "LEARNING VIA RETRACING"

We wish to examine the effects of the proposed adaptive scheduling of truncation (Section 3.3). From Figure 3b, we observe that the original *CCWM* is outperformed by Dreamer on the Hopper Stand task, probably due to the large degree of "irreversibility" of the task comparing to the others such that the naive representation learning by enforcing "learning via retracing" leads to suboptimal training.

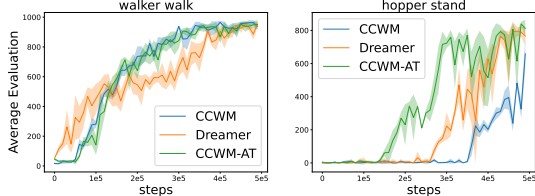

From Figure 5, we observe that augmenting *CCWM* with the proposed flexible truncation schedule yields significant performance increase on the Hopper-Stand task, with consistently better sample efficiency than both Dreamer and the standard *CCWM*. For tasks with less degree of "irreversibility", such as Walker-Walk (Figure 5 right), we do not observe significant improvement by the introduction of adaptive truncation since the amount of negative impacts were already minimal in the original settings. Similar patterns are observed across the other tasks (no noticeable improvement except for the Hopper-Stand/Hop tasks). We extend our discussion on adaptive truncation in Appendix C.

Figure 5: Evaluation of Adaptive Truncation on tasks with varying degrees of "irreversibility". Minimal improvement is observed on tasks with low degree of "irreversibility" (walker walk in the left panel); whereas significant improvements are observed on tasks with high degree of "irreversibility" (hopper stand in the right panel).

## 6 DISCUSSION

We proposed "learning via retracing", a novel representation learning method for RL problems that utilises the temporal cycle-consistency of the transition dynamics in addition to the predictive reconstruction in the temporally forward direction. We introduce *CCWM*, a concrete model-based instantiation of "learning via retracing" based on generative dynamics modelling. We empirically show that *CCWM* yields improved performance over state-of-the-art model-based and model-free methods on a number of challenging continuous control benchmarks, in terms of both the sample-efficiency and the asymptotic performance. We also show that "learning via retracing" supports stronger generalisability and more accurate long-range predictions, hence stronger planning, both adhere nicely to our intuition and predictions. We note that "learning via retracing" is strongly affected by the degree of "irreversibility" of the task. We propose an adaptive truncation scheme for alleviating the negative impacts caused by the "irreversible" transitions, and empirically show the utility of the proposed truncation mechanism in tasks with a large degree of "irreversibility".

Hippocampal replay has long been thought to play a critical role in model-based decision-making in humans and rodents (Mattar and Daw, 2018; Evans and Burgess, 2019). Recently, Liu et al. (2021) showed how reversed hippocampal replays are prioritised due to its utility in non-local (model-based) inference and learning in humans. *CCWM* can be interpreted as an immediate model-based instantiation of the reversed hippocampal replay. Similar to intuitions from the computational neuroscience literature (Penny et al., 2013), we also find that "learning via retracing" brings a number of merits comparing to its counterparts that only uses forward rollouts. In addition to improved sample efficiency and asymptotic performance, we show that *CCWM* also supports stronger generalisability and extends the planning horizon, indicating that stronger model-based inferences are obtained. Moreover, as discussed previously, "learning via retracing" can enable the learning of reversible "controllable" internal states of the agent even when external features (e.g. location) are irreversible. Such representation could be combined with task-specific global contextual information/representation provided by an independent system to facilitate stronger policy learning. The independent system could be supported by the hippocampal area, where reversed hippocampal "replay" of the allocentric "cognitive map" could in turn drive reverse replay of egocentric sensory or motoric representations.

## ACKNOWLEDGEMENT

C.Y. is supported by the DeepMind studentship funded through UCL CDT in Foundational Artificial Intelligence. N.B. is supported by the Wellcome Trust. The authors would like to thank Maneesh Sahani, Peter Dayan, Jessica Hamricks, and the anonymous reviewers for helpful comments and discussions.

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

# A    PSEUDOCODE FOR *CCWM*

The pseudocode for *CCWM* training is shown in Algorithm 1. In the current instance, we show the full algorithm with the augmentation of adaptive truncation (Section 3.3). Given the adaptive truncation, we need to slightly modify the loss function in Eq. 8.

$$\mathcal{L}(\theta, \psi, \zeta) = \frac{1}{NT} \sum_{n=1}^{N} \sum_{\tau=1}^{T} \left[ \mathcal{L}_{\text{ELBO}}(O_\tau^n; \theta, \psi) + \lambda M_\tau^n \mathcal{L}_{\text{retrace}}(\tilde{z}_\tau^n, \check{z}_\tau^n; \psi, \zeta) \right], \tag{9}$$

where $M_\tau^n$ is the entry $\tau$ of the mask vector $M^n$ with values in $\{0, 1\}$, which we describe in Algorithm 1.

---

**Algorithm 1** *Cycle-Consistency World Model (CCWM)*

---

1: **Given:**   policy $\pi(a|s)$, encoders $q_\phi(O)$, $q_{\psi_1}(z'|z, a)$, $q_{\psi_2}(z'|z, a, q_\phi(O'))$, decoder $q_\theta(O|z)$, reverse action approximator $\rho_\zeta(z, z')$, latent horizon $K$, average loss $\mathcal{L}_{\text{avg}} = 0$, counter $c = 0$, adaptive truncation threshold, $\eta$, adaptive truncation sliding window size, $S$, current estimate of the Q-function, $Q_\gamma$, *CCWM* parameters: $\Theta = \{\phi, \psi_1, \psi_2, \theta, \zeta\}$.
2: **Input:** Batch of $N$ sampled trajectories of length $T$, $\{\mathbf{O}_{t_n:t_n+T}^n, \mathbf{a}_{t_n:t_n+T}^n\}_{n=1}^N$.
3: **for** $n = 1$ **to** $N$ **do**
4:      $O = \{\mathbf{O}_{t_n+k}^n\}_{k=1}^T$, $a = \{\mathbf{a}_{t_n+k}^n\}_{k=0}^{T-1}$.
5:      **for** $k = 1$ **to** $T - K$ **do**
6:          Compute prior $\hat{z}_{k+1:k+K}$ and posterior estimates $\tilde{z}_{k+1:k+K}$ using Eq. 5
7:          Compute the sliding window average, $[\bar{Q}_1, \dots, \bar{Q}_{K-S}]$, where $\bar{Q}_i = \frac{1}{S} \sum_{s=1}^S Q_\gamma(\tilde{z}_{i+s}, a_{i+s})$
8:          Compute the step-wise difference, $\Delta = [|\frac{\bar{Q}_2 - \bar{Q}_1}{\bar{Q}_1}|, \dots, |\frac{\bar{Q}_{K-S} - \bar{Q}_{K-S-1}}{\bar{Q}_{K-S-1}}|]$
9:          Compute the adaptive truncation mask, $M \in \{0, 1\}^K$, where $M_k = 1$ if $\vee_{j=k-S}^k (\Delta_j < \eta)$, and 0 otherwise
10:          Compute retraced states $\check{z}_{k:k+K}$ using Eq. 6
11:          Compute the latent model loss $\mathcal{L}_k$ using Eq. 9.
12:          Update $\mathcal{L}_{\text{avg}} \leftarrow \frac{c}{c+1} \mathcal{L}_{\text{avg}} + \frac{1}{c+1} \mathcal{L}_k$.
13:          Increment counter, $c \leftarrow c + 1$.
14:      **end for**
15: **end for**
16: Compute the gradient, $\nabla_\Theta \mathcal{L}_{\text{avg}}$.
17: Update the model parameters, $\Theta \leftarrow \Theta + \alpha \nabla_\Theta \mathcal{L}_{\text{avg}}$.

---

# B    MODEL-FREE INSTANTIATION OF "LEARNING VIA RETRACING"

As mentioned in Section 3, "learning via retracing" admits many degrees of freedom in its implementation. *CCWM* provides one such instantiation under the model-based RL setting, here we provide an alternative model based on "learning via retracing" under the model-free RL setting. The graphical illustration of the model-free instantiation is shown in Figure 6.

Visual inspection indicates the high similarity between the graphical models of the model-free version and the model-based version (CCWM), but there are essential differences. Similar to PlayVirtual (Yu et al., 2021), due to the model-free nature of the model, we no longer requires further supervisory signals obtained from "learning via retracing" to contribute to training of the dynamics model, hence we are free to employ an independent "reversed" dynamics model (denoted by the red arrows in the reversed direction in Figure 6) for performing the retracing operations. Moreover, given the independent "reversed" dynamics model, we no longer requires approximation of the "reversed" actions, hence removing the necessity of using $\rho$ as in Figure 2b, and we only need to use the ground-truth forward actions for the retracing operations. The learned representation in this case would benefit the downstream model-free RL agent since the resulting state representation is efficient for the prediction of future states. We note a key difference between our model-free instantiation of learning via retracing and PlayVirtual (Yu et al., 2021), that we have consistently employed probabilistic

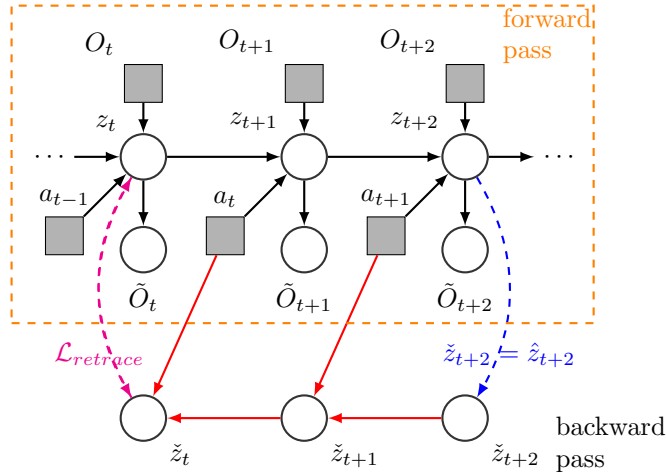

Figure 6: Graphical illustration of a model-free instantiation of "learning via retracing". The forward model is a state-space model and is trained in a generative fashion under the variational principles. The retracing operations are now performed with a separate "reversed" dynamics model (indicated by the red arrows). Given the independent "reversed" dynamics model, we can use the same action as in the forward model for retracing, removing the necessity of using the "reversed" action approximator.

models over deterministic models for modelling the embedding and latent transitions, which naturally provides posterior predictive uncertainty that can be used for various downstream tasks, such as exploration (Osband et al., 2016).

Here we stick with the general architectural choice of using a sequential state-space model for the forward dynamics model as in CCWM, but the "reversed" dynamics model can be chosen to be deterministic and trained discriminatively jointly with the entire model. Note that Figure 6, like CCWM, only describes one of many possible instantiations of "learning via retracing", we leave further investigation to future work.

## C  FURTHER DISCUSSION ON TRUNCATION AND THE DEGREE OF "IRREVERSIBILITY"

In Section 3.3, we introduced the adaptive truncation as a general-purpose method for improving the performance of *CCWM* in continuous control tasks. On a high level, we choose to remove the transitions from model/representation learning through "learning via retracing" whenever large changes in the Q-values occur, hence the "irreversibility" of the transitions depend on the local continuity of the corresponding Q-values. Our argument is largely based on Theorem 1 from Gelada et al. (2019), which we re-iterate below for consistency.

**Theorem 1** *For an MDP $\mathcal{M} = \langle \mathcal{S}, \mathcal{A}, \mathcal{R}, \mathcal{P}, \gamma \rangle$ and the corresponding DeepMDP $\bar{\mathcal{M}} = \langle \bar{\mathcal{S}}, \mathcal{A}, \bar{\mathcal{R}}, \bar{\mathcal{P}}, \gamma \rangle$ (see Section 2.2 in Gelada et al. (2019) for detailed definition of DeepMDP), let $d_{\bar{\mathcal{S}}}$ be a metric over $\bar{\mathcal{S}}$, $\phi : \mathcal{S} \to \bar{\mathcal{S}}$ be the embedding function, $L_{\bar{\mathcal{R}}}^{\infty} = \sup_{s \in \mathcal{S}, a \in \mathcal{A}} |\mathcal{R}(s, a) - \bar{\mathcal{R}}(\phi(s), a)|$ and $L_{\bar{\mathcal{P}}}^{\infty} = \sup_{s \in \mathcal{S}, a \in \mathcal{A}} W(\phi \mathcal{P}(\cdot|s, a), \bar{\mathcal{P}}(\cdot|\phi(s), a))$ be the global loss function (for training the DeepMDP, where $W(\cdot, \cdot)$ is the Wasserstein distance). For any $K_{\bar{V}}$-Lipschitz policy $\bar{\pi} \in \bar{\Pi}$, the representation $\phi$ guarantees that for all $s_1, s_2 \in \mathcal{S}$, and $a \in \mathcal{A}$,*

$$|Q^{\bar{\pi}}(s_1, a) - Q^{\bar{\pi}}(s_2, a)| \leq K_{\bar{V}} d_{\bar{\mathcal{S}}}(\phi(s_1), \phi(s_2)) + 2 \frac{L_{\bar{\mathcal{R}}}^{\infty} + \gamma K_{\bar{V}} L_{\bar{\mathcal{P}}}^{\infty}}{1 - \gamma} \qquad (10)$$

The proof of Theorem 1 can be found in Appendix A.2 in Gelada et al. (2019). The high-level interpretation for Theorem 1 is that the learned representation space should admit the situation where two states with different values collapse onto the same representations. Theorem 1 tells us that given a good representation (embedding function), the absolute difference between the Q-values

of two latent states should lower-bound the distance (some metric over the representation space) between the representations of the two states (up to some additive and scalar multiplicative constants). Hence whenever there is a large change in the Q-values of between the adjacent states in a trajectory, there must also be a large jump in the representation space, which we interpret as an "irreversible" transition. The latter interpretation can be substantiated with the visualisation of the low-dimensional embeddings of the learned representations at different stages in Figure 7 and Figure 8. Figure 7 shows the TSNE embeddings of a trajectory taken by a successfully trained agent in the Cheetah Run task (we focus on the left figure in both Figure 7(a) and (b), please refer to the full discussion of Figure 7 in Appendix E). We see that the resulting TSNE-embeddings of the representation along a trajectory show that the magnitude of the transitions in the representation space is consistently low, hence respecting the temporal locality of the states along the trajectory. However, in Figure 8, where we show the TSNE-embeddings of the representation along a trajectory in an under-trained CCWM agent in the Hopper Stand task ($\sim 2 \times 10^5$ training steps), we observe clear clusters of the state representations along the same trajectory. Moreover, the clustered structure also shows locality with respect to the temporal ordering (along the trajectory), showing that even in tasks such as Hopper Stand, which we view as the more "irreversible" task, a large proportion of all transitions allows retracing operations, which can contribute as additional samples for the training of representation and the dynamics model through "learning via retracing". The quality of the overall representation learning is dependent on the correct identification of the "irreversible" transitions (large jumps between adjacent states in Figure 8), which necessitates the usage of the adaptive truncation approach to identify and remove the "irreversible" transitions from corrupting the learning process. These evidence further substantiate our proposal of the overall framework of "learning via retracing" and the adaptive truncation technique.

We note that as the agent is undergoing a sequence of consecutive "irreversible" transitions, such as when the walker agent is falling down shown in Figure 1(b) of the main paper, the transitions the agent goes through are "irreversible" albeit having similar Q-values (all leading to the inevitable fallen down state). In order to deal with such "continuity" in the "irreversibility" of the transitions along the same trajectory, the adaptive truncation is introduced by detecting the changes in the average Q-values over a sliding window instead of the single Q-values, which allows a delayed detection of the sudden (and extended) changes in the state values along the same trajectory (the degree of delay depends on the choice of the threshold of adaptive truncation and the sliding window size, see Section 3.3 and Algorithm 1 for more details). Hence upon the "delayed" identification of the "irreversible" transition at state $s_t$, we remove the state as well as a certain number of states preceding $s_t$ (i.e., $\{s_{t-\tau}, s_{t-\tau+1}, \dots, s_t\}$) from "learning via retracing". We note that the choice of the sliding window size, $S$, threshold, $\eta$ and the number of preceding states to remove, $\tau$, are all chosen based on hyperparameter tuning at this stage. We leave the automatic determination of the parameters associated with adaptive truncation to future work.

We additionally propose an intermediate approach for ruling out the "irretraceable" states from "learning via retracing", based on fixed scheduling of truncating out the final proportions of each collected episode (but still use these samples for representation learning in the forward direction). In many episodic environments, (e.g., LunarLander; Brockman et al. (2016)), each episode terminates upon either successful completion of the task or failing the task (e.g., crashing in LunarLander). For the failure cases, which is more often during the early phase of training, even the transitions leading to the failure states might be considered "irreversible" (falling android; Figure 1b). Hence early truncation of each episode is able to rule out many such "irreversible" transitions from contributing to representation/model learning. We additionally propose annealing schedules for the truncation proportion as given the training of the controller, successful completion will be more frequently encountered comparing to the failure cases.

## D    IMPLEMENTATION DETAILS

We implement the latent transition dynamics model of *CCWM* using the RSSM (Hafner et al., 2019), utilising a Gated Recurrent Unit (GRU) as the core component (Chung et al., 2014) in combination with multi-layer perceptrons (MLP). The RSSM is used for modelling the predictive inferences, $q_{\psi_1}(z_{t+1}|z_t, a_t)$ and $q_{\psi_2}(z_{t+1}|z_t, a_t, e_{t+1})$, from Eq. 2. The context embedding function is given by a convolutional encoder that deterministically embeds the high-dimensional observation into a context vector. The dimension of the latent space is set to equal 32, there is low sensitivity of

the performance with respect to the dimension of the latent embedding. The generative function $q_\theta(O_t|z_t)$ is implemented with the combination of an MLP and a deconvolution network (Zeiler et al., 2010), outputting the means of the Gaussian distributions for each pixel value of the decoded observation. The reward distribution $\hat{\mathcal{R}}(r|z)$ is modelled by an MLP, outputting a univariate Gaussian distribution. The supervision of the model learning is based on the ELBO and the bisimulation metric as described in Section 3.

In the implementation of our policy agent, A3C, we implement the critic (value network) using an MLP, with latent state inputs. The value network models a Gaussian distribution with means and variances. The policy network also describes a Gaussian distribution, with means and variances parameterised by an MLP. We used distributed actors for interacting with the environment for more efficiency collection of sampled episodes as in (Mnih et al., 2016). The value function is optimised by maximising the probability of the "ground-truth" values estimated by the lambda return (Sutton and Barto, 2018). The policy network is updated using policy gradient estimated based on generalised advantage estimation (GAE) ()schulman2015high.

We use the Adam optimiser for updating the parameters for the model, value function, and policy networks (Kingma and Ba, 2014).

The specific configurations for the neural network implementations is summarised in Table 2. Note that the activation functions for all non-output layers are assumed to be ReLU activation function unless otherwise stated.

All neural network implementation are carried out using TensorFlow (Abadi et al., 2015) and TensorFlow Distributions (Dillon et al., 2017).

For the actual training, the batch size is chosen to be $64$, and all sampled trajectories are taken to be $50$ timesteps long. Greedy evaluation is performed every $10^4$ training steps. The reported evaluation scores are averaged values over $5$ random seeds. We adopt the same scheme for setting the action repeats equal to $2$ across all tested environments as in (Hafner et al., 2020a). The parameter $\lambda$ controlling the weights of the retrace auxiliary loss in Eq. 8 is set to $1.0$. The discounting factor for the expected value function is set to $0.99$.

The predictive reconstruction (Figure 4, Figure 10) is performed by firstly applying the dynamics model to the first $5$ frames of the trajectory, and starting from the posterior latent estimates at step $5$, the latent predictions are performed given solely the action inputs, and the predicted latent estimates are then decoded back into the observable space. To ensure fair comparison, as well as to demonstrate that *CCWM* improves model learning beyond solely sample efficiency, the model of the baseline Dreamer is provided further training, until the training steps is twice the training steps of *CCWM* and the final performance reaches a comparable score of *CCWM*. Specifically, for both the walker walk and cheetah run tasks, the Dreamer model used for reconstruction is provided with $2 \times 10^6$ training steps.

For the implementation details of the generalisability experiments in Section 5.3, we manually changed the corresponding settings of the agent in the task-dependent XML file (see e.g., `https://github.com/deepmind/dm_control/blob/master/dm_control/suite/cheetah.xml`). We choose the semantically interpretable physical quantities that we hypothesise to alter the underlying task MDP, including changing mass from $14$ to any one of the list: $[6, 8, 10, 20]$, changing the friction constants from $(0.4, 0.1, 0.1)$ to any element of the list: $\{(i,j,k)|i \in \{0.6, 0.8\}, j \in \{0.4, 0.5\}, k \in \{0.3, 0.4\}\}$, changing the stiffness constants from $8$ to any of the element in $\{2, 6, 10\}$. We directly evaluate (without any further training) the trained CCWM and Dreamer on the original task structure on the tasks corresponding to each of the above changes and any combination of the individual changes. The change in reward setting is merely a constant reward reshaping across all transitions, and should not result in any difference in the task dynamics, hence the evaluations, and we use this as the sanity check and a baseline.

Note that in Eq. 7, we follow (Zhang et al., 2020) to replace the 1-Wasserstein distance with the 2-Wasserstein distance, since the 2-Wasserstein distance between two Gaussians has closed-form expression:

$$W_2(\mathcal{N}(\mu_i, \Sigma_i), \mathcal{N}(\mu_j, \Sigma_j)) = ||\mu_i - \mu_j||_2^2 + ||\Sigma_i^{1/2} - \Sigma_j^{1/2}||_{\mathcal{F}}^2 \qquad (11)$$

where $|| \cdot ||_{\mathcal{F}}$ is the matrix Frobenius norm. This admits analytical computation instead of having sample from the joint distribution hence reducing the variance of loss function.

| COMPONENT | ATTRIBUTE | VALUE |
|---|---|---|
| RSSM | INTERNAL STATE DIMENSION | 256 |
| | MLP LAYER 1 UNITS | 256 |
| | MLP LAYER 2 UNITS | 64 |
| EMBEDDING FUNCTION | CONV LAYER 1 NUMBER OF FILTERS | 32 |
| | CONV LAYER 1 KERNEL SIZE | 4 |
| | CONV LAYER 2 NUMBER OF FILTERS | 64 |
| | CONV LAYER 2 KERNEL SIZE | 4 |
| | CONV LAYER 3 NUMBER OF FILTERS | 128 |
| | CONV LAYER 3 KERNEL SIZE | 4 |
| | CONV LAYER 4 NUMBER OF FILTERS | 256 |
| | CONV LAYER 4 KERNEL SIZE | 4 |
| | FINAL OUTPUT OPERATION | CONCATENATION |
| GENERATIVE MODEL | MLP LAYER 1 NUMBER OF UNITS | 1024 |
| | DECONVOLUTION LAYER 1 NUMBER OF FILTERS | 128 |
| | DECONVOLUTION LAYER 1 KERNEL SIZE | 5 |
| | DECONVOLUTION LAYER 2 NUMBER OF FILTERS | 64 |
| | DECONVOLUTION LAYER 2 KERNEL SIZE | 5 |
| | DECONVOLUTION LAYER 3 NUMBER OF FILTERS | 32 |
| | DECONVOLUTION LAYER 3 KERNEL SIZE | 6 |
| | DECONVOLUTION LAYER 4 NUMBER OF FILTERS | 3 |
| | DECONVOLUTION LAYER 4 KERNEL SIZE | 6 |
| REWARD MODEL | MLP LAYER 1 NUMBER OF UNITS | 512 |
| | MLP LAYER 2 NUMBER OF UNITS | 512 |
| | MLP LAYER 3 NUMBER OF UNITS | 1 |
| VALUE MODEL | MLP LAYER 1 NUMBER OF UNITS | 512 |
| | MLP LAYER 2 NUMBER OF UNITS | 512 |
| | MLP LAYER 3 NUMBER OF UNITS | 512 |
| | MLP LAYER 4 NUMBER OF UNITS | 1 |
| POLICY MODEL | ALL FULLY CONNECTED LAYERS ACTIVATION | ELU |
| | MLP LAYER 1 NUMBER OF UNITS | 512 |
| | MLP LAYER 2 NUMBER OF UNITS | 512 |
| | MLP LAYER 3 NUMBER OF UNITS | 512 |
| | MLP LAYER 4 NUMBER OF UNITS | 512 |
| | MLP LAYER 5 NUMBER OF UNITS | 2 |
| ADAM OPTIMISER | LEARNING RATE FOR MODEL LEARNING | $6 \times 10^{-4}$ |
| | LEARNING RATE FOR VALUE MODEL | $8 \times 10^{-5}$ |
| | LEARNING RATE FOR POLICY MODEL | $8 \times 10^{-5}$ |

Table 2: Configurations for the neural network implementations of *CCWM*.

For our implementation of the adaptive truncation technique (Section 3.3), the associated parameters that require pre-specification are the detection threshold, $\eta$, adaptive truncation sliding window size, $S$. There are also the optional parameters, the number of states preceding the detected changes to be omitted from "learning via retracing" (which is set to equal to $S$ by default, see Line 9 in Algorithm 1 and Section 3.3), $\tau$, and the number of initial steps to be omitted from adaptive truncation (due to under-training of the critic, which is set to 0 by default), $\xi$. The default values for the parameters we used for the empirical evaluation shown in Figure 5 are: $\eta = 0.10$, $S = 10$, $\tau = 5$, $\xi = 1 \times 10^5$. The hyperparameters are determined through standard grid search, and we leave the automatic determination of the parameters associated with adaptive truncation to future work.

The python implementation of *CCWM* can be found at `https://github.com/changmin-yu/CCWM_code`.

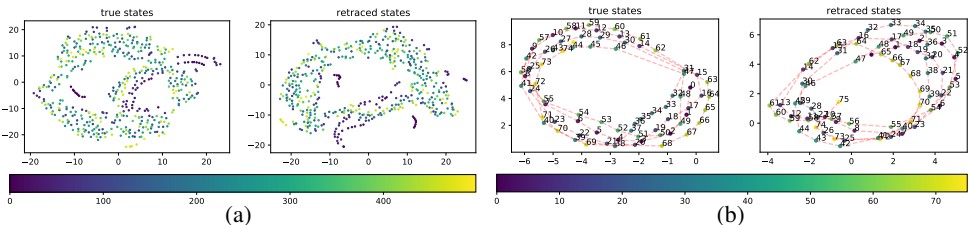

Figure 7: **Visualisation of similarity between the ground-truth states and retraced states using tSNE.** The two-dimensional embeddings for a 500-step trajectory (a) and a 76-step subtrajectory (b) is shown. Color of the scatters indicates the temporal ordering of the states within the trajectories.

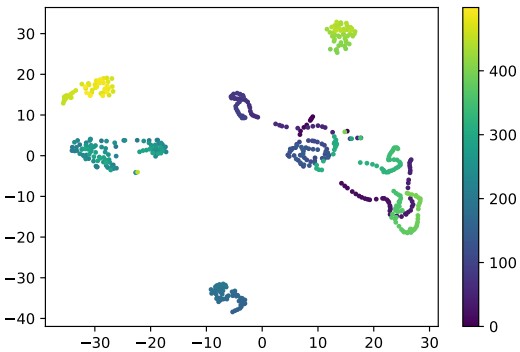

Figure 8: **Visualisation of state representations along the same trajectory in Hopper Stand task.** We show the TSNE-embeddings of the state representations over a trajectory of length 500 steps in Hopper Stand task, with a standard CCWM agent (without adaptive truncation) that has received $2 \times 10^5$ training steps. Clear clustered structured can be observed in the visualisation, which indicates: (a) the existence of "irreversible" transitions, and (b) the majority of the transitions are "reversible", hence "learning via retracing" should still apply to facilitate improved sample efficiency.

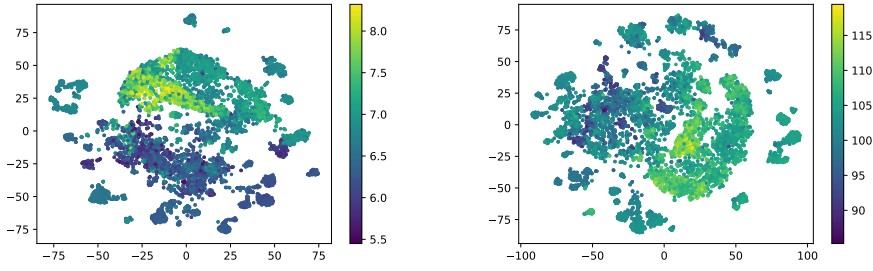

Figure 9: **Visualisation of the structure of the state representations with respect to the Q-values.** We show the TSNE-embeddings of the state representations of 10000 randomly sampled states (from the cached replay buffer) of CCWM (left) and Dreamer (right) in the Walker Walk task. Visual inspections show that CCWM leads to more disentangled state representations with respect to the state-values, hence potentially supports stronger generalisability (Higgins et al., 2016).

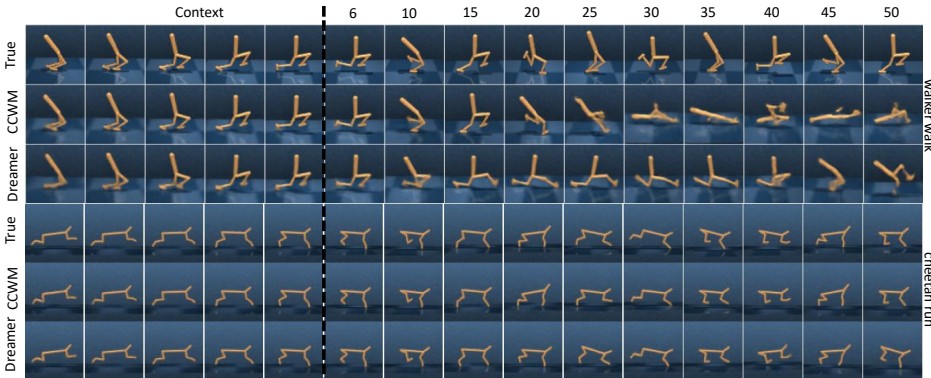

Figure 10: **Qualitative comparison of long-range predictive reconstruction of *CCWM* and Dreamer.** Predictive rollouts over 45 time-steps given the actions are computed using the representation models. Comparing to Dreamer, *CCWM* consistently generates more accurate predictive reconstructions over longer time spans. *CCWM* generates accurate predictions up to $\sim 18$ steps on Walker task, and $\sim 35$ steps on Cheetah task; Dreamer generates accurate predictions up to $\sim 8$ steps on Walker task, and $\sim 15$ steps on Cheetah task (evaluated over 5 randomly sampled trajectories).

# E    FURTHER INVESTIGATION OF THE EFFECTS OF "LEARNING VIA RETRACING" IN THE SAMPLE EFFICIENCY OF LEARNING

In order to demonstrate that "learning via retracing" indeed brings additional "valid" supervision signals for model learning, we need to examine whether the retracing operation is correctly learned. In Figure 7 we show the t-SNE embeddings of the true and retraced latent states over a trajectory of 500 steps of the trained agent on the cheetah run task (Van der Maaten and Hinton, 2008). We observe a similar dominant ring structure in both embeddings. This matches our intuition, since during running, the simulated cheetah repeats its states periodically to resume running at a high speed. This is more clearly observed from a sub-trajectory towards the end of the entire episode in Figure 7b, where most states are densely distributed on both sides of the ring and more loosely distributed on the paths connecting the two sides for both the original and retraced states. Moreover, during the early stage of the episode (darker dots in Figure 7a), the agent starts to accelerate from stationary and the states of the agent are periodic in nature but differs from the states when the agent is running at full speed. Such early-phase structure is correctly captured by the retracing operation. Collectively these evidence support that the retracing is correctly learned hence provides useful supervision signals for model training.

# F    LONG-RANGE PREDICTION RECONSTRUCTION

In Figure 10, we show the complete 45-step predictive reconstructions for *CCWM* and Dreamer on "walker walk" and "cheetah run" tasks. We see that *CCWM* is able to retain accurate predictions over the entire 45 predictive time-steps in the cheetah task, whereas Dreamer fails to do so. By looking more carefully at Figure 10, latent rollouts in the embedding space learned by *CCWM* yield less accurate predictions for the "irrecoverable" states (e.g., when the agent is falling, see Section 5.4 for further discussion). This property of the learned representation enables "task-aware planning", i.e., the behaviour learning agent is trained to acquire more precise control policy such that the "irrecoverable" states that might hinder the overall learning process are minimally encountered, leading to improved efficiency and quality of learning.

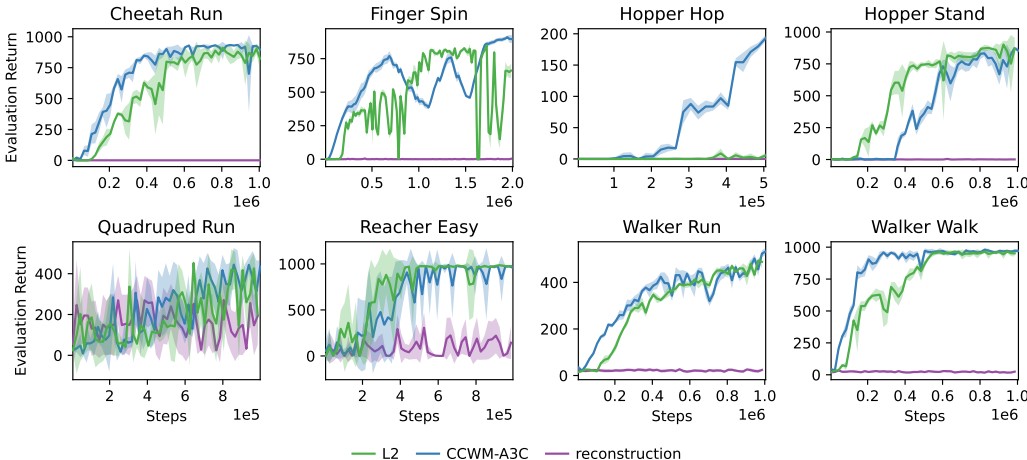

Figure 11: **Full ablation studies of the retracing loss function.** *CCWM* with bisimulation metrics as the retracing loss function consistently outperforms the alternatives using the $L2$ and reconstruction retracing error.

| CHANGED COMPONENTS | CCWM (MEAN $\pm 1$ S.D.) | DREAMER (MEAN $\pm 1$ S.D.) | P-VALUE (3 S.F.) | SIGNIFICANT? ($\alpha = 0.01$) |
|---|---|---|---|---|
| R | $630.40 \pm 6.49$ | $629.00 \pm 47.01$ | $5.22 \times 10^{-1}$ | NO |
| R + M | $635.37 \pm 40.12$ | $597.43 \pm 87.68$ | $7.84 \times 10^{-2}$ | NO |
| R + F | $643.58 \pm 9.29$ | $649.27 \pm 3.96$ | $9.76 \times 10^{-1}$ | NO |
| R + S | $634.80 \pm 10.92$ | $646.07 \pm 7.78$ | $9.98 \times 10^{-1}$ | NO |
| R + M + F | $\mathbf{628.07 \pm 36.95}$ | $468.82 \pm 94.73$ | $\mathbf{7.27 \times 10^{-6}}$ | YES |
| R +M + S + F | $\mathbf{641.89 \pm 28.67}$ | $562.58 \pm 93.91$ | $\mathbf{3.97 \times 10^{-3}}$ | YES |

Table 3: Evaluation of trained *CCWM* and Dreamer on the ability of zero-shot transfer in cheetah run tasks with different configurations. (R: Reward; M: Mass; F: Friction; S: Stiffness.)

## G  FULL ABLATION STUDIES ON THE RETRACE LOSS FUNCTION

In Figure 11, we show the full ablation studies of the retracing loss function in *CCWM*. We observe that the bisimulation metric retracing loss function consistently outperforms the other alternatives (L2 and reconstruction error). This result conforms with our hypothesis that cycle-consistency supervision given bisimulation metrics poses a stronger constraint on the learning of the representation, leading to a learned representation that better respect the cycle-consistency properties of the environment.

## H  FULL RESULTS OF ZERO-SHOT TRANSFER

Please refer to Table 3 for full results on the zero-shot transfer experiment. Note that the $R$-changes only rescale the reward function by a constant, hence should not count towards a transfer task.

