# OpenReview forum: "Learning State Representations via Retracing in Reinforcement Learning"
_ICLR.cc/2022/Conference — ICLR 2022 Poster_

### Official Review · Reviewer_VSAG · 2021-10-29

**Correctness:** 3
**Technical Novelty And Significance:** 3
**Empirical Novelty And Significance:** 3
**Recommendation:** 8
**Confidence:** 3

**Main Review:**

Pros:

- Interesting technique to retrieve more supervisory signals from the data in RL
- Mathematically sound approach, with clear figure 2 to visually understand.
- Extensive experiments to evaluate the idea and interesting ablations and investigations described.
- Extensive details of algorithm and implementation + code is available.

##########################################################################

Cons:

- A few inconsistencies in the paper that make me think that there were extra parts that have been cut out to reach the page limit without reformatting:
    - Section 5.3: Details of modifications to the environment are said to be found in Appendix A, but Appendix A describes implementation details and not the environment.
    - Appendix C, last sentence. Model-free instantiation is shown in Figure ?? Figure doesn't exist.

##########################################################################

Questions during rebuttal period:

- Address/clarify cons above.
- How do you select the predefined threshold to find the sudden changes in the value function for your adaptive scheduling?
- I can imagine from Table 1, that the components that you change in the cheetah-run environment are related to Reward, Mass, Friction, Stiffness. I don't understand how changing the reward would affect evaluating an episode, as it doesn't change the dynamic behavior of the physics simulator and so the actual trained model shouldn't have any problem in evaluating.
- In Section 5.4, you write that the curve for hopper-stand in Figure 3b for CCWM lacks the adaptive truncation of irreversible states. Does this mean that all plots in Figure 3b for CCWM are trained on all transitions, with no truncation? Please clarify, as I imagined that the experimental evaluation of your method included both contributions (CCWM+adaptive truncation).
- In algorithm 1: I don't understand in the input, why in ${O^n_{t_n:t_n+K}, a^n_{t_n:t_n+K}}$ the $t$ has a subscript $n$, which should represent the n-th sample in the batch. Also, I think there is a typo and the subscript should be to $t+T$ and not $t+K$.
- In figure 8 (from Appendix F), you explain how CCWM yields less accurate predictions for the irrecoverable states, but to me the first row (true trajectory for Walker), doesn't look like an irrecoverable trajectory.

##########################################################################

Typos/formatting:
- Section 3.3, first paragraph: "Here we propose an approaches" -> "Here we propose an approach"
- None of the citations in Appendix A are in brackets anymore.
- Appendix A: "outputting the means the Gaussian distributions" -> "outputting the means of the Gaussian distributions"
- Appendix D, second paragraph: "empirical evaluation" -> "empirically evaluate"

**Summary Of The Paper:**

This paper proposes "learning via retracing" as an approach to learn state representations, through matching a trajectory both in the forward and in the backward direction. This paper then introduces Cycle-Consistency World Model (CCWM) which is a model-based RL algorithm which learns through retracing. This method is sample efficient and provides good state representations for predicting the future and generalization. Furthermore, it proposes a value-based approach to identify "irreversible" transitions.


**Summary Of The Review:**

Overall, I am for accepting. I like the idea and I think it's a novel approach for state representations in RL. My major concern is about the clarity of the paper, as written above. Hopefully the authors can address my concern in the rebuttal period.

---

> ### Author Response · Authors · 2021-11-11
> **Thank you for your reviews, please see below for our replies to your questions/concerns (Part I)**
>
> We thank the reviewer for the thoughtful comments. Below we address each of the raised questions/concerns.
>
> [Implementation details of the generalisability experiments]
>
> We thank the reviewer for pointing out the omission of implementation details of the generalisability experiments in Table 1, which we have now included in Appedix A in the updated version of the manuscript. We apologise for the omission at the first instance.
>
> [Missing illustrative demonstration of model-free version of "learning vis retracing"]
>
> We apologise for the omission of the illustrative figure of the model-free version of "Learning via Retracing", which we have now provided in Appendix C of the update manuscript. Visual inspection indicates the high similarity between the graphical models of the model-free version and the model-based version (CCWM), but there are key differences. On a high-level, similar to PlayVirtual (Yu, et al. 2021), due to the model-free nature, it is no longer priority to use the additional "retracing" samples to train the same dynamics model (although we still can), hence we propose to use independent forward and "reversed" dynamics model where the "reversed" dynamics model is chosen to be a discriminative model and trained in a supervised learning fashion. Moreover, since we use an independent "reversed" dynamics model (which is not technically a dynamics model, rather a plain regression model), we no longer require $\rho$, the reversed action approximator. We wish to emphasise that we introduce "learning via retracing" as a general framework for representation learning, but model-based methods enable additional benefits of improved dynamics model learning, hence we choose CCWM as our main agent for empirical evaluation of the effects of "learning via retracing". We thank the reviewer for pointing out the missing of the figure.
>
> [Selection of threshold for adaptive truncation]
>
> The threshold of the changes in the Q-values (e.g., $|Q_{2}-Q_{1}|/Q_{1}|$) is set by standard grid-search, we tried different threshold values in the list [0.05, 0.10, 0.15, 0.20, 0.25, 0.30], and determine the best value in terms of The optimal setting of the threshold is 0.10 for both the Hopper Stand and Walker Walk tasks, we found 0.10 is generally a good setting for 7 out of 8 tasks, yielding improved or comparable performance comparing to standard CCWM agent.
>
> [Changing components of Cheetah Run task in the generalisability experiment]
>
> The reviewer is correct about the general changes we made to the task. Indeed, change to the rewards is superfluous as this is supposed to be a uniform reward shaping, which should not lead to any difference in terms of evaluation comparing to the original task. We use this as a sanity check that things work as we expected. We will consider removing the inclusion of reward changes in the final version upon acceptance.
>
> [No adaptive truncation in all training in Figure 3(b)]
>
> The reviewer is correct about that all training shown in Figure 3(b) is the standard CCWM, i.e., without augmentation of the adaptive truncation. Only Figure 5 shows the performance of standard CCWM + adaptive truncation. The reasons for such demonstration is that in Figure 3, the main goal is to show the utility of "learning via retracing" alone. After providing sounding empirical evidence that "learning via retracing" is indeed beneficial for the overall learning, we then, in Figure 5, show the utility of adaptive truncation in terms of its ability of minimising the negative impacts brought by the "irreversible" transitions.
>
> [Further clarifications regarding Algorithm 1]
>
> In the training of CCWM, when we sample trajectory data from the replay buffer, instead of sampling the entire trajectories, we sample a sub-trajectory from each trajectory, hence we also randomly sample the starting point of each sub-trajectory in each batch, hence the $t_{n}$ notation.
>
> We thank the reviewer for pointing out the typo in algorithm 1, which we have changed in the updated version of the manuscript.
>
> [Please see below for the remaining replies due to word limits]

---

> > ### Author Response · Authors · 2021-11-11
> > **Thank you for your reviews, please see below for our replies to your questions/concerns (Part II)**
> >
> > [Less accurate predictions for "irrecoverable" transitions in Figure 8]
> >
> > We expect the predictive reconstruction of CCWM to be less accurate when CCWM has made an "irrecoverable" transition, instead of when the ground-truth trajectory shows the occurrence of an "irrecoverable" transition. We note that one reason such inconsistency with the ground-truth trajectory might occur is the predictive uncertainties in the forward rollout sampling process, where the uncertainties accumulate exponentially with rollout length since we use the predictive prior with no access to the ground-truth observations. We note that such inconsistency is present in all world-model type agents with sequential state-space models, and we see that CCWM still outperforms Dreamer in terms of predictive rollout quality.
> >
> > [Typos]
> >
> > We thank the reviewer for pointing out the typos, we have addressed them in the updated version of the manuscript.
> >
> > We again thank the reviewer for the reviews. We hope the above replies help to clarify the raised concerns, and we are happy to answer any further questions/comments you may have.

---

> > > ### Comment · Reviewer_VSAG · 2021-11-19
> > > **updated score**
> > >
> > > Thanks for your reply. I updated my score accordingly.

---

> > > > ### Author Response · Authors · 2021-11-22
> > > > **Thanks for raising the score**
> > > >
> > > > We wish to thank the reviewer for raising the score, and we are delighted to see that our replies have helped with clarifying the concerns.

---

### Official Review · Reviewer_dZvG · 2021-11-02

**Correctness:** 3
**Technical Novelty And Significance:** 3
**Empirical Novelty And Significance:** 3
**Recommendation:** 5
**Confidence:** 3

**Main Review:**

The strengths of the paper involve solid experiments and well-structured writing. However, I think some details are not illustrated clearly, so that I have the following questions:
1. Is the representation learning part jointly trained with RL algorithms or pretrained?
2. In the $L_{retrace}$, why retracing samples $\check{z}$ are from the variational predictive distribution but forward samples $\tilde{z}$ are from the variational posterior distribution?
3. How the “reversed” action policy $p_{\eta} $ is learned? by $L_{retrace}$?
4. The proposed method for dealing with “irreversibility” is too intuitive. I think it might fail in heavily irreversible environments. Why the proposed truncation method is not involved in your full algorithm in the appendix?
5.  As you mentioned CCWM could be combined with any existing RL algorithms, why not try one or two to show that the new representation learning loss could actually improve the performance. I think the performance of CCWM is comparable with Dreamer given the huge additional computational complexity.


**Summary Of The Paper:**

This paper proposes a self-supervised approach for learning the state representation of RL tasks. The main contribution apart from prior works is to involve additional 'retracing' trajectory samples in representation training, which are trained by minimizing the propensity between retraced samples and forward posterior samples. They also proposed an intuitive way to mitigate the irreversibility in RL dynamics.

**Summary Of The Review:**

The proposed representation learning method for RL is interesting, which involves addtional retracing samples for training. However, more explanations are needed to make it clear. I expect to see more elaborations.

---

> ### Author Response · Authors · 2021-11-11
> **Thank you for your reviews, please see below  for our replies to your questions/concerns (Part I)**
>
> We thank the reviewer for the thoughtful comments. Below we address each of the raised questions/concerns.
>
> [Is the representation learning part jointly trained with RL algorithms or pretrained?]
>
> The representation learning (and the dynamics model learning) is jointly trained with the RL algorithms. The overall training procedure follows closely with other standard world-model type agents such as Dreamer (Hafner, et al. 2019) and SLAC (Lee, et al. 2019).
>
> [Why retracing samples are from the predictive prior instead of the posterior?]
>
> We allow the agent to perform the retracing operation similar to "Dreaming" in training the RL agent with the learned model, i.e., predictions are based on only the predictive prior rather than the predictive posterior with the injection of the observation information, such that stronger representation learning can be achieved since less information is provided for the retracing operations. We use the predictive posterior samples in predictive reconstruction supervision since this is standard from the variational free energy formulation (ELBO) that the Monte Carlo estimate of the objective function is based on the samples from the posterior distributions.
>
> [How the reversed policy is learned?]
> There is no "reversed" policy, rather the reversed action approximator $\rho$ outputs the approximate "reversed" actions given the inputs of $s_{t}$ and $s_{t+1}$, and the reversed action approximator $\rho$ is trained jointly with the overall model learning, which is from $L_{retrace}$.
>
> [Inclusion of adaptive truncation in the full algorithm and application of "learning via retracing in heavily "irreversible" environments.]
>
> The full algorithm in appendix B describes the algorithm we used for performing the empirical evaluations in Figure 3, i.e., the original version of CCWM. We have provided the new pseudocode for the algorithm augmented with adaptive truncation in appendix B in the updated manuscript.
>
> The empirical studies in the main paper are based on continuous control tasks, which are generally not perfectly "reversible". We have shown that despite the "irreversibility" inherent in such tasks, the improved sample-efficiency and stronger representation learning brought by the direct application of "learning via retracing" generally outweigh the negative impacts brought by the "irreversible" transitions (Figure 3, 4, 7, Table 1). Moreover, with the introduced truncation technique (Section 3.3) that attempts to avoid the "irreversible" transitions, we demonstrate that CCWM augmented with adaptive truncation yields state-of-the-art performance on tasks with strong "irreversibility" (such as Hopper tasks) where the plain CCWM failed to outperform the baselines. Most tasks admit "reversible" transitions to some extent, hence adaptive truncation which identifies "irreversible" transitions based on the continuity of the trained representation space appears to be effective,and any densely irreversible tasks will just benefit less from "learning via retracing".
>
> [Combination with other RL algorithms and computational complexity of CCWM]
> A main goal of our empirical studies is to demonstrate the improved sample-efficiency and asymptotic performance brought by augmenting existing models with "learning via retracing". Hence in order to demonstrate our arguments, we choose to maintain our implementation of CCWM as closely as possible to our main baseline agent, Dreamer, to establish fair comparisons. If we choose to use, e.g., SAC, as the RL algorithm, it would raise questions regarding whether the improvement is brought by the introduction of "learning via retracing" or by the more powerful RL algorithm. Moreover, from Figure 3, we can clearly see that CCWM outperforms Dreamer on 5 tasks (Cheetah Run, Finger Spin, Hopper Hop, Walker Run, Walker Walk), and CCWM is comparable with Dreamer on 2 tasks (Quadruped Run, Reacher Easy), and is outperformed by Dreamer on 1 task (Hopper Stand). With the additional augmentation of adaptive truncation (Figure 5b), we showed that CCWM diminished the negative impacts brought by the "irreversible" transitions and is able to outperform Dreamer on the Hopper Stand task by a large margin where it failed to yield comparable or better performance previously. Hence we believe our statement that CCWM outperforms Dreamer in terms of both sample-efficiency and asymptotic performance is well-justified.
>
> [Please see below for the remaining replies due to word limits]

---

> > ### Author Response · Authors · 2021-11-11
> > **Thank you for your reviews, please see below for our replies to your questions/concerns (Part II)**
> >
> > CCWM, given our motivation that we wish to train a dynamics model that is coherent with the cycle-consistency property of the underlying MDP transition dynamics, we use the same dynamics model for both the forward and "reversed" transitions, removing the necessity of using an independent "reversed" dynamics model (as in PlayVirtual (Yu, et al. 2021)), hence we expect to see minimal increase in the improved computational complexity. For instance, in the Walker Walk experiment, there is a 13\% increase in wall-clock training time for 1e6 steps using CCWM over using Dreamer, and there is a 5.2\% increase in the number of model parameters of CCWM comparing to Dreamer, whereas we see that there is at least 25\% improvement in terms of sample efficiency (Figure 3 in the main paper), where CCWM reaches asymptotic performance at \~3e5 steps and Dreamer reaches similar asymptotic performance at over 4e5 steps. Moreover, the amount of increase in computational complexity (in terms of both wall-clock time and memory) is uniform across the presented tasks. We hence conclude that the improvement brought by "learning via retracing" in CCWM outweighs the increase in the computational complexity.
> >
> > We again thank the reviewer for the reviews. We hope the above replies help to clarify the raised concerns, and we are happy to answer any further questions/comments you may have.
> >
> > References:
> >
> > [1] Hafner, Danijar, et al. "Dream to control: Learning behaviors by latent imagination." arXiv preprint arXiv:1912.01603 (2019).
> >
> > [2] Lee, Alex X., et al. "Stochastic latent actor-critic: Deep reinforcement learning with a latent variable model." arXiv preprint arXiv:1907.00953 (2019).

---

### Official Review · Reviewer_NEVM · 2021-11-02

**Correctness:** 2
**Technical Novelty And Significance:** 3
**Empirical Novelty And Significance:** 2
**Recommendation:** 3
**Confidence:** 4

**Main Review:**

- The motivation for the cycle consistency is not really discussed. Why would enforcing cycle consistency improve representation learning?
- In equation 7: can you explain the choice of the KL distance between two rewards? How is the 2-wassertstein distance estimated from data?
- The "model-free instantiation of learning via retracing" (appendix C) is not clear. In particular, the paper states that there is a graphical illustration "shown in Figure" but there is no reference to any figure.
- In Figure 3, can you clarify why some of the baselines are straight lines? Can you provide more details than this explanation: 'We report the asymptotic scores for the model-free algorithms due to the large gap in sample efﬁciency comparing to the model-based methods."
- Section 3.3: The prediction of the irreversibility of a transition is not very clear. How is a "sudden change in the value function" a good predictor of the irreversibility of the transition?
- In the conclusion, it is written "We empirically show that CCWM yields improved performance over state-of-the-art model-based and model-free methods on a number of challenging continuous control benchmarks, in terms of both the sample efﬁciency and the asymptotic performance." However, CCWM is only compared to dreamer and yields to comparable results in most cases? (the other baselines are only given as "asymptotic scores" and they have a direct access to the state?)

Minor comments:
- The paper formalizes an observation space, O that is "high-dimensional, due to either redundant information or simply because only visual inputs are available". Can you clarify whether that is the POMDP setting? If it is not the POMDP setting, it might be worth clarifying as this is the usual denomination used in POMDP.
- The paper is based on the open source implementation of Hafner et al. Is there an open source implementation of the modifications presented in this paper?

**Summary Of The Paper:**

The paper investigates a self-supervised approach for learning the state representation in RL tasks. In addition to the predictive (reconstruction) supervision in the forward direction, the authors include a “retraced” transitions for representation/model learning, by enforcing the cycle-consistency constraint between the original and retraced states. The authors claim that this facilitates stronger representation learning and improve upon the sample efﬁciency of learning. As it is not always possible to find such a cycle consistency between two states and for counteracting the negative impacts brought by the “irreversible” transitions, the authors add a novel adaptive “truncation” mechanism.

**Summary Of The Review:**

The motivation for the contribution is not very clear and there are a few unclear elements in the formalization (e.g. sudden change in the value function for the prediction of the irreversibility) .

---

> ### Author Response · Authors · 2021-11-11
> **Thank you for your review, please see below for our replies to your questions/concerns (Part I)**
>
> We thank the reviewer for the thoughtful comments. Below we address each of the raised questions/concerns.
>
> [Motivation of "Learning via Retracing"]
>
> In Section 1 of the main paper, we discussed the motivations for "learning via retracing". The main motivation is that by utilising the temporal cycle-consistency of the trajectory data, we can, by enforcing the cycle-consistency constraints in the overall model learning, achieve additional supervisory signals for the training of both the representations and the dynamics model without any further interactions with the environment/simulator compared to existing methods that uses forward models (twice as much in tasks with perfect "reversibility"), hence improving the overall sample-efficiency of learning. Moreover, we provide an intuitive example in Figure 1 (a) in the main paper, giving a high-level interpretation of why "learning via retracing" might support stronger generalisation. Namely, given the simulated retracing steps, the key states for successful completion of the task (e.g., bottleneck states in the maze-navigation task) can be quickly and accurately inferred, hence supporting stronger transferability across different tasks.
>
> [KL between reward distributions and computation of the Wasserstein distance]
>
> In the original definition of the bisimulation metric proposed in Ferns, et al. 2011, the difference between the rewards are quantified as the L1 distance between the rewards. However, in our implementations, since we formulate reward prediction probabilistically (as a Gaussian distribution), we cannot only compute the difference between the expected rewards, rather we need to compute the dissimilarity based on the sufficient statistics of the Gaussian distributions, which involve both the mean and variance. Hence the KL-divergence is a better metric for quantifying the differences between the reward predictions than the L1 distance.
>
> Note that in CCWM, we follow Zhang, et al. 2020, to replace the 1-Wasserstein distance in the original definition from Ferns, et al. 2011 with the 2-Wasserstein distance, since the 2-Wasserstein distance admits the close-form expression for Gaussian distributions. Please see the exact expression in Appendix A (Eq. 9) and the accompanying discussions in the updated version of the manuscript.
>
> [Missing Figure of the model-free version of "Learning via Retracing"]
>
> We apologise for omitting the illustrative figure of the model-free version of "Learning via Retracing", which we have now provided in Figure 6 in Appendix C of the update manuscript. Visual inspection indicates the high similarity between the graphical models of the model-free version and the model-based version (CCWM), but there are key differences. On a high-level, similar to PlayVirtual (Yu, et al. 2021), due to the model-free nature, it is no longer priority to use the additional "retracing" samples to train the same dynamics model (although we still can), hence we propose to use independent forward and "reversed" dynamics model where the "reversed" dynamics model is chosen to be a discriminative model and trained in a supervised learning fashion. Moreover, since we use an independent "reversed" dynamics model (which is not technically a dynamics model, rather a plain regression model), we no longer require $\rho$, the reversed action approximator. We wish to emphasise that we introduce "learning via retracing" as a general framework for representation learning, but model-based methods enable additional benefits of improved dynamics model learning, hence we choose CCWM as our main agent for empirical evaluation of the effects of "learning via retracing". We thank the reviewer for pointing out the omission of the figure.
>
> [Straight lines for the model-free baselines in empirical evaluations]
>
> In Figure 3, we compare CCWM with a number of existing methods. One reason for only showing straight lines (asymptotic performance) of the model-free methods is that the model-free methods receive the proprioceptive state (i.e., the ground-truth state) information instead of visual observations as inputs, leading to unfair comparisons. Moreover, we show the asymptotic performance of the model-free methods because despite the improved sample-efficiency, many model-based methods fail to perform comparably to the model-free methods in terms of asymptotic performance (Janner, et al. 2019). Hence for the empirical evaluations in Figure 3, our main purpose is to show that CCWM, the model-based instantiation of "learning via retracing", yields improved sample-efficiency whilst still achieving comparable asymptotic performance to the model-free methods receiving true state information as inputs.
>
> [Please see below for the remaining replies due to word limits]

---

> > ### Author Response · Authors · 2021-11-11
> > **Thank you for your review, please see below for our replies to your questions/concerns (Part II)**
> >
> > [Q-values as indicators for adaptive truncation]
> >
> > In Section 3.3 of the main paper, we introduced adaptive truncation as a measure for counteracting the negative impacts brought by "irreversible" transitions. Briefly speaking, whenever there is a large change in the Q-values in the trajectory, we do not apply "learning via retracing" to these transitions as we assume there is a high chance that such transitions are "irreversible". This argument is largely based on Theorem 1 from Gelda, et al. 2019, which on a high level, states that given a good representation space, two states with different Q-values should not collapse onto the same representation. An immediate consequence of the theorem is that the absolute difference between the Q-values of two latent states should lower-bound the distance (some metric over the representation space) between the representations of the two states (up to some additive constants). This tells us that if there is a large change in the Q-values of the states in a sequence, there must also be a large jump in the representation space, which we interpret as the "irreversible" transitions (this latter interpretation can be substantiated with the empirical observations from Figure 7, which shows that upon successful training in tasks with less "irreversibility", such as Cheetah tasks, the transitions in the representation space respect temporal locality). Please see the appendix D in the updated submission for for detailed discussions.
> >
> > [Conclusion of empirical performance]
> >
> > One main goal of the paper is to empirically show that "learning via retracing" indeed yields improved performance, hence we apply the method to the SOTA model-based world-model agent, Dreamer. And from our empirical evaluations, we confirm that CCWM, which is essentially a Dreamer agent augmented with "learning via retracing", indeed induces the improvement expected from our analysis and motivations. From Figure 3, we can clearly see that CCWM outperforms Dreamer on 5 tasks (Cheetah Run, Finger Spin, Hopper Hop, Walker Run, Walker Walk), and CCWM is comparable with Dreamer on 2 tasks (Quadruped Run, Reacher Easy), and is outperformed by Dreamer on 1 task (Hopper Stand). With the additional augmentation of adaptive truncation (Figure 5b), we showed that CCWM outperforms Dreamer on the Hopper Stand task by a large margin (and adaptive truncation shows smaller improvements for the highly "reversible" tasks, such as Walker walk, see Figure 5(a)). Hence we believe our statement that "We empirically show that CCWM yields improved performance over state-of-the-art model-based and model-free methods on a number of challenging continuous control benchmarks, in terms of both the sample efﬁciency and the asymptotic performance." is well-justified. We hope the above discussions resolve the reviewer's concerns about the justification of our conclusion regarding empirical performance, and we are happy to proceed with further discussions if still found unclear.
> >
> > [High-dimensional O\_t...]
> >
> > We confirm that in the work we indeed formulate the problem under the MDP setting over the representation space, however as we only receive the image inputs instead of the proprioceptive inputs (containing the ground-truth state information), the descriptions are indeed similar to the POMDP settings. We thank the reviewer for pointing out that such statement might leads to misunderstanding, please see the new description in Section 2.1 of the updated manuscript.
> >
> > [Open-Source Implementation]
> >
> > In the attached supplementary materials, we provide the codes for the implementation of CCWM. The code is currently not available as open-source due to the double-blind nature of the review process, but will be made available to the public upon acceptance.
> >
> > We again thank the reviewer for the reviews. We hope the above replies help to clarify the raised concerns, and we are happy to answer any further questions/comments you may have.
> >
> > References:
> >
> > [1] Ferns, N., Panangaden, P., \& Precup, D. (2011). Bisimulation metrics for continuous Markov decision processes. SIAM Journal on Computing, 40(6), 1662-1714.
> >
> > [2] A. Zhang, R. McAllister, R. Calandra, Y. Gal, and S. Levine. Learning invariant representations for reinforcement learning without reconstruction. ArXiv, abs/2006.10742, 2020.
> >
> > [3] Yu, Tao, et al. "Playvirtual: Augmenting cycle-consistent virtual trajectories for reinforcement learning." arXiv preprint arXiv:2106.04152 (2021).
> >
> > [4] Janner, Michael, et al. "When to trust your model: Model-based policy optimization." arXiv preprint arXiv:1906.08253 (2019).

---

> > > ### Comment · Reviewer_NEVM · 2021-11-16
> > > **Not fully convinced by the rebuttal on two key points**
> > >
> > > Dear authors,
> > > Thanks for your detailed reply. The two main concerns that I have described earlier and that remain are the following:
> > > - The prediction of the irreversibility of a transition is still not very clear, despite your general answer about it and the individual answers (as the issue was also pointed out by Reviewer HLp2 and also to some extend by Reviewer dZvG). When considering your example from Figure 1, all the states where the agent is anyway falling are "irreversible" in the sense that there is no action that allows coming back to the previous state, yet they have a quite similar value function. How does the algorithm handles such transitions?
> > > - The motivation for the cycle consistency is described as "improving representation learning", however there is nothing that can directly back this up (as compared for instance to other model-based learning losses?), except the improved generalization that you show on some benchmarks. I do not see any theoretical justification or specific visualizations/experiments that directly support the claim about the improved representation learning.

---

> > > > ### Author Response · Authors · 2021-11-18
> > > > **Thank you for your extended reply, we address the raised two key points below.**
> > > >
> > > > We thank the reviewer for the fast response to our replies, below we address the two remaining concerns.
> > > >
> > > > [Similar value functions during falling down]
> > > >
> > > > The reviewer points out that as the agent is undergoing a sequence of consecutive "irreversible" transitions, such as when the walker agent is falling down, as shown in Figure 1(b) of the main paper, the transitions the agent goes through are "irreversible" albeit having similar Q-values (all leading to the inevitable fallen down state). We are aware of such situations and outlined measures for dealing with it in our description of the adaptive truncation mechanism in Section 3.3 (we apologise if this was not clear). We detect sudden changes in the average Q-value over a sliding window, which allows a delayed detection of sudden (and extended) changes in the state values along the trajectory (the degree of delay depends on the choice of the sliding window size, see Section 3.3 and Algorithm 1 in Appendix B for more details). Hence upon the "delayed" identification of the "irreversible" transition at state $s_{t}$, we remove the state as well as a number of states preceding $s_{t}$ (i.e., $\{s_{t-\tau}, s_{t-\tau+1}, \dots, s_{t}\}$) from "learning via retracing". We note that the choice of the sliding window size, $S$, threshold, $\eta$ and the number of preceding states to remove, $\tau$, are all chosen based on hyperparameter tuning at this stage (please see the second last paragraph in Appendix for further details of the hyparameter setting). We leave the automatic determination of the parameters associated with adaptive truncation to future work.
> > > >
> > > > [Improved Representation Learning]
> > > >
> > > > The reviewer questions whether learning via retracing does produce an improvement in  representation learning, as we claim, and how we could visualise or demonstrate this.
> > > >
> > > > First we note that CCWM is an augmentation of the learning of the  dynamics model within Dreamer  with "learning via retracing", by constraining the temporal cycle-consistency between the retraced and original latent states (i.e. augmentation of the learning of the representation of the task dynamics). No additional changes are made to other aspects of the overall architecture, such as the actor-critic agent, the network architecture of the dynamics model, etc. A better dynamics model enables the construction of more task-aware latent state variables, hence leading to improved representation learning. Of course, a better dynamics model will lead to better policy learning and better performance – this is the definition of a better dynamics model as its only function is to aid policy learning.
> > > >
> > > > So, because learning via retracing only affects representation learning, and this results in improved performance, we have demonstrated that learning via retracing improves representation learning. We note that the improvement in performance includes sample efficiency, asymptotic performance, generalisability, and more accurate and longer model-based rollouts, and these all result from our augmentation of learning the dynamics model. We hope you now see the logic of our statement and have tried to make this conclusion more clear in the paper.
> > > >
> > > > In terms of visualising the improvement in representation learning we showed that the CCWM agent shows stronger generalisability than Dreamer (Table 1), and that CCWM generates more accurate predictions over a longer time span than Dreamer (in terms of qualitative similarities between the predictive reconstructions and the ground-truth observations, shown in Figure 4 and Figure 11). In Figure 10 of the updated manuscript of the paper, we show the TSNE embeddings of the representations of 10000 randomly sampled states (from the replay buffer) given CCWM and Dreamer in the Walker Walker task. Visual inspections show a noticeable increase in the degree of disentanglement of the TSNE embeddings using CCWM than using Dreamer, hence showing that more semantically sensible state representations are learned than that of Dreamer, and potentially leading to stronger generalisation (Higgins, et al. 2016), which is further substantiated and confirmed by our generalisability experiment (Table 1, Section 5.3 in the main paper).
> > > >
> > > > We again thank the reviewer for the extended replies. We hope the above discussions and associated empirical evidence have resolved the reviewer's questions regarding the details of the adaptive truncation techniques and our conclusion of the improvement in representation learning. We are happy to extend the discussion if the reviewer still finds our discussion unclear or have any further questions/concerns.
> > > >
> > > > References:
> > > >
> > > > [1] Higgins, Irina, et al. "beta-vae: Learning basic visual concepts with a constrained variational framework." (2016).

---

### Official Review · Reviewer_HLp2 · 2021-11-03

**Correctness:** 4
**Technical Novelty And Significance:** 3
**Empirical Novelty And Significance:** 3
**Recommendation:** 6
**Confidence:** 3

**Main Review:**

Strength: The cycle-consistency constraints could provide extra supervision signal (beyond forward predictive loss) for learning state representations and transition models, without additional interaction with environment. It also improves the representation learning performance (including the zero-shot transfer capability and long-range predictions) by obtaining better latent state inference.

Weakness: The concept of cycle-consistency constraints have also been considered in the PlayVirtual work under model-free setting. Therefore, more thorough discussion about the novel contribution relative to PlayVirtual should be included.

Comments:
- The authors may need to discuss how frequent such cycle-consistency constraints appear in practical applications. For example, it is more common in navigation-type applications? Listing more practical application settings in this regard and discuss them would help justify how widely applicable the proposed approach is.
- Why the continuity detection on action-value function could detect the irreversible states? It seems that there is not much discussion on this point in Section 3.3. It is crucial to clarify it.
- VirtualPlay should also be included as a baseline in the experiment section in order to show which approach best exploits cycle-consistency.


**Summary Of The Paper:**

This paper considers state representation learning problem in deep RL. It exploits the cycle-consistency supervision and develops a “learning via retracing” approach. Such supervision signals are generally inherent in existing data and does not need additional interaction with environment, which leads to better sample efficiency. Learning from predictive supervision from temporally forward and backward directions reveals information from both the future and past to the target state, leading to more accurate latent state inference. In particular, the paper proposes the Cycle-Consistency World Model (CCWM) along with practical considerations (e.g., adaptive truncation to remove irreversible states), under the model-based framework based on generative dynamics modeling (CCWM).

**Summary Of The Review:**

The paper proposes CCWM, a learning via retracing method, based on cycle-consistency constraint. It leads to better sample efficiency and final performance by learning better state representations. Need more clarification about its novel contribution relative to a recent work (VirtualPlay), which considers cycle-consistency learning in model-free RL setting.

---

> ### Author Response · Authors · 2021-11-11
> **Thank you for your review, please see below for our replies to your questions/concerns (Part I)**
>
> We thank the reviewer for the thoughtful comments. Below we address each of the raised questions/concerns.
>
>
> [Novelty with respect to PlayVirtual.]
>
> In the current version of the paper, we explicitly recognise that PlayVirtual is a model-free instantiation of the general concept of "learning via retracing" (Section 4, Paragraph 2), which also utilises the temporal cycle-consistency inherent in the trajectory data for improved sample-efficiency of representation learning and the overall reinforcement learning. While we recognise the similarities, it is also worth making clear the differences between CCWM and PlayVirtual by inspecting the overall "learning via retracing" framework. Specifically, there are three "major" elements for the instantiation of "learning via retracing":
> - model-free or model-based;
> - computation of the "reversed" actions;
> - implementation of the "reversed" dynamics model;
>
> It is clear that CCWM and PlayVirtual lie on the opposite ends of the spectrum of the possible instantiations of "learning via retracing" based the above "major" elements. Namely, CCWM is a model-based agent whereas PlayVirtual is model-free; CCWM utilises a "reversed" action approximator, $\rho$, for computing the "reversed" actions, whereas PlayVirtual takes the original forward actions as the inputs to the retracing operations; CCWM uses the same dynamics model for latent transitions in both the forward and "reversed" directions, whereas PlayVirtual maintains a separate Backward Dynamics Model (BDM), similar to SPR (Schwarzer, et al. 2021).
>
> These "major" differences reveals the conceptual novelty of CCWM with respect to PlayVirtual. PlayVirtual, as a model-free method, mainly targets the faster acquisition of the state representations; whereas in CCWM, as a model-based method, by utilising the same dynamics model for both forward and "reversed" transitions, we hope to
> % embed the inductive bias that the majority of the transitions are "retraceable" (i.e., there exists valid actions to retrace to $s_{t}$ form $s_{t+1}$) into the training of the dynamics model, hence leading to
> produce a learned dynamics model that is more coherent with the ground-truth environmental transition rules.
>
> There are a number of further differences, such as CCWM uses a generative latent transition model, and PlayVirtual uses a discriminative transition model that is trained in a supervised fashion; CCWM utilises the bisimulation metric for implementing the temporal cycle-consistency constrains, and PlayVirtual uses the "projection metric" proposed in the SPR paper (Schwarzer, et al. 2021); PlayVirtual uses data augmentation techniques in model learning whereas CCWM does not. Such architectural differences does not lead to changes in the underlying design principles of the models, hence we categorise them as the "minor" differences.
>
> [Applicability of "Learning via Retracing"]
>
> The empirical studies in the main paper are based on continuous control tasks, which are generally not perfectly "reversible". We have shown that despite the "irreversibility" inherent in such tasks, the improved sample-efficiency and stronger representation learning brought by the direct application of "learning via retracing" generally outweigh the negative impacts brought by the "irreversible" transitions (Figure 3, 4, 7, Table 1). Moreover, with the introduced truncation technique (Section 3.3) that attempts to avoid the "irreversible" transitions, we demonstrate that CCWM augmented with adaptive truncation yields state-of-the-art performance on tasks with strong "irreversibility" (such as Hopper tasks) where the plain CCWM failed to outperform the baselines. Most tasks admit "reversible" transitions to some extent, hence adaptive truncation which identifies "irreversible" transitions based on the continuity of the trained representation space appears to be effective (see below and the updated submission),and any densely irreversible tasks will just benefit less from "learning via retracing". % In the worst case scenario, where the task admits complete "irreversibility" across all possible transitions (e.g., sky-diving), truncation will stop all training signals from "learning via retracing", leading to standard representation learning with only forward models (e.g., Dreamer).
>
> [Please see below for the remaining replies due to word limits]

---

> > ### Author Response · Authors · 2021-11-11
> > **Thank you for your review, please see below for our replies to your questions/concerns (Part II)**
> >
> > [Q-values as indicators for adaptive truncation]
> >
> > In Section 3.3 of the main paper, we introduced adaptive truncation as a measure for counteracting the negative impacts brought by "irreversible" transitions. Briefly speaking, whenever there is a large change in the Q-values in the trajectory, we categorise such transitions as "irreversible". This argument is largely based on Theorem 1 from Gelda, et al. 2019, which on a high level, states that given a good representation space, two states with different Q-values should not collapse onto the same representation. An immediate consequence of the theorem is that the absolute difference between the Q-values of two latent states should lower-bound the distance (some metric over the representation space) between the representations of the two states (up to some additive constants). This tells us that if there is a large change in the Q-values between the adjacent states in a trajectory, there must also be a large jump in the representation space, which we interpret as the "irreversible" transitions. The interpretation that the distance over the representation space conveys the degree of "irreversibility" of the transitions can be substantiated with the empirical observations from Figure 8 and 9 in the updated version of the manuscript, which shows that upon successful training in tasks with less "irreversibility", such as Cheetah tasks, the transitions in the representation space respect temporal locality, whereas in tasks with stronger "irreversibility", such as Hopper Stand, the states along the same trajectory tend to fall into well-separated clusters, where the within-cluster states are close in terms of the temporal ordering (along the trajectory). Further discussions can be found in the appendix D and the new Figure 9 in the updated submission.
> >
> > [Adding VirtualPlay as a Baseline algorithm in empirical evaluations]
> >
> > In our above discussion regarding the novelty of CCWM with respect to PalyVirtual, we discussed a number of major and minor differences between the two agents. Some of these differences make PlayVirtual an inapproriate baseline for CCWM. For instance, PlayVirtual utilises data augmentation, which CCWM does not, and many recent works have shown that data augmentation is an important factor in the improvement of sample efficiency. The main aim of our empirical studies in Section 5 is to show that "learning via retracing" indeed provide the expected benefits such as improved sample-efficiency and stronger generalisability, as per our analysis and motivations. Hence in order to maximally demonstrate the utility of "learning via retracing", we implement CCWM as closely as possible to Dreamer (e.g., same architectures for the forward model and the actor-critic agent). By contrast, PlayVirtual is based on SPR, which integrates a number of techniques that are absent in CCWM. Despite that we recognise that PlayVirtual is an important related work to the general framework of "learning via retracing" and CCWM, but given the different architectural choices, we believe adding PlayVirtual as a baseline agent would not add further support to the arguments we are trying to make in the paper, hence we will respectfully decline this suggestion at the current instance.
> >
> > We again thank the reviewer for the reviews. We hope the above replies help to clarify the raised concerns, and we are happy to answer any further questions/comments you may have.
> >
> > References:
> >
> > [1] Schwarzer, Max, et al. "Data-efficient reinforcement learning with self-predictive representations." arXiv preprint arXiv:2007.05929 (2020).
> >
> > [2] Gelada, Carles, et al. "Deepmdp: Learning continuous latent space models for representation learning." International Conference on Machine Learning. PMLR, 2019.

---

### Author Response · Authors · 2021-11-11
**Summarised Reply and Updated Manuscript (Part I)**

We wish to thank all the reviewers for their efforts and thoughtful comments. Here we wish to draw the reviewers' attention to the common questions/concerns that are raised by more than one reviewers, and the associated replies and changes we made to the submitted manuscript.

[Q-values as indicators for adaptive truncation]

In Section 3.3 of the main paper, we introduced adaptive truncation as a measure for counteracting the negative impacts brought by "irreversible" transitions. Briefly speaking, whenever there is a large change in the Q-values in the trajectory, we categorise such transitions as "irreversible". This argument is largely based on Theorem 1 from Gelda, et al. 2019, which on a high level, states that given a good representation space, two states with different Q-values should not collapse onto the same representation. An immediate consequence of the theorem is that the absolute difference between the Q-values of two latent states should lower-bound the distance (some metric over the representation space) between the representations of the two states (up to some additive constants). This tells us that if there is a large change in the Q-values between the adjacent states in a trajectory, there must also be a large jump in the representation space, which we interpret as the "irreversible" transitions. The interpretation that the distance over the representation space conveys the degree of "irreversibility" of the transitions can be substantiated with the empirical observations from Figure 8 and 9 in the updated version of the manuscript, which shows that upon successful training in tasks with less "irreversibility", such as Cheetah tasks, the transitions in the representation space respect temporal locality, whereas in tasks with stronger "irreversibility", such as Hopper Stand, the states along the same trajectory tend to fall into well-separated clusters, where the within-cluster states are close in terms of the temporal ordering (along the trajectory).

In appendix D of the updated manuscript, we now provide a more detailed discussion for our intuition and evidence for using the Q-values are indications of "irreversibility" detection in the adaptive truncation technique. Moreover, we have included a new Figure 9 in the appendix, which together with Figure 8, provide us intuitive visual examples for explaining why "learning via retracing" could be generally helpful in terms of providing additional high-quality supervisory signals for representation learning, and how the adaptive truncation mechanism could further improve the applicability of "learning via retracing".

[Missing Figure of the model-free version of "Learning via Retracing"]

We apologise for omitting the illustrative figure of the model-free version of "Learning via Retracing", which we have now provided in Figure 6 in Appendix C of the update manuscript. Visual inspection indicates the high similarity between the graphical models of the model-free version and the model-based version (CCWM), but there are key differences. On a high-level, similar to PlayVirtual (Yu, et al. 2021), due to the model-free nature, it is no longer priority to use the additional "retracing" samples to train the same dynamics model (although we still can), hence we propose to use independent forward and "reversed" dynamics model where the "reversed" dynamics model is chosen to be a discriminative model and trained in a supervised learning fashion. Moreover, since we use an independent "reversed" dynamics model (which is not technically a dynamics model, rather a plain regression model), we no longer require $\rho$, the reversed action approximator. We wish to emphasise that we introduce "learning via retracing" as a general framework for representation learning, but model-based methods enable additional benefits of improved dynamics model learning, hence we choose CCWM as our main agent for empirical evaluation of the effects of "learning via retracing". We thank the reviewers for pointing out the omission of the graphical illustration.

[Please see below for the remaining replies due to word limits]

---

> ### Author Response · Authors · 2021-11-11
> **Summarised Reply and Updated Manuscript (Part II)**
>
> [Conclusion of improved empirical performance]
>
> We are happy to see the reviewers are paying attention and raising thoughtful questions regarding the empirical evaluations, here we summarise the concerns and questions and address them.
>
> Given our empirical evaluations, we confirm that CCWM, which is essentially a Dreamer agent augmented with "learning via retracing", indeed induces the improvement expected from our analysis and motivations. From our main evaluation figure (Figure 3), we can clearly see that CCWM outperforms Dreamer on 5 tasks (Cheetah Run, Finger Spin, Hopper Hop, Walker Run, Walker Walk), and CCWM is comparable with Dreamer on 2 tasks (Quadruped Run, Reacher Easy), and is outperformed by Dreamer on 1 task (Hopper Stand). With the additional augmentation of adaptive truncation (Figure 5b), we showed that CCWM outperforms Dreamer on the Hopper Stand task by a large margin (and adaptive truncation shows smaller improvements for the highly "reversible" tasks, such as Walker walk, see Figure 5(a)). Hence we believe our statement that "We empirically show that CCWM yields improved performance over state-of-the-art model-based and model-free methods on a number of challenging continuous control benchmarks, in terms of both the sample efﬁciency and the asymptotic performance." is well-justified.
>
> One main goal of our empirical studies is to show that "learning via retracing" indeed yields improved performance, in terms of sample-efficiency, asymptotic performance, stronger generalisability, and interpretability. Hence in order to maximally demonstrate the advantage of "learning via retracing", we choose to maintain our implementation of CCWM as closely as possible to our main baseline agent, Dreamer, to establish fair comparisons. Additional changes to the model architecture would raise questions regarding whether the improvement is brought by the introduction of "learning via retracing" or by the power of the additional techniques (e.g., different RL agent, data augmentation, etc.).
>
> CCWM, given our motivation that we wish to train a dynamics model that is coherent with the cycle-consistency property of the underlying MDP transition dynamics, we use the same dynamics model for both the forward and "reversed" transitions, removing the necessity of using an independent "reversed" dynamics model (as in PlayVirtual (Yu, et al. 2021)), hence we expect to see minimal increase in the improved computational complexity. For instance, in the Walker Walk experiment, there is a 13\% increase in wall-clock training time for 1e6 steps using CCWM over using Dreamer, and there is a 5.2\% increase in the number of model parameters of CCWM comparing to Dreamer, whereas we see that there is at least 25\% improvement in terms of sample efficiency (Figure 3 in the main paper), where CCWM reaches asymptotic performance at \~3e5 steps and Dreamer reaches similar asymptotic performance at over 4e5 steps. Moreover, the amount of increase in computational complexity (in terms of both wall-clock time and memory) is uniform across the presented tasks. We hence conclude that the improvement brought by "learning via retracing" in CCWM outweighs the increase in the computational complexity.
>
> We hope the reviewers find the above discussions and the individual replies useful for providing a clearer picture of the general framework of "learning via retracing" and the specific model-based instantiation, CCWM. We welcome any further questions/comments/concerns, and are happy to engage in further discussions. We again thank the reviewers for their time and efforts in the reviewing process.

---

### Author Response · Authors · 2021-11-23
**We thank the reviewers for the reviews and look forward to further feedbacks**

Dear reviewers,

We thank you again for the valuable comments and your time and efforts during the review and rebuttal process. We look forward to hearing from you with respect to any further feedback.

If you find the responses satisfactory, we hope you might view this as a sufficient reason for further raising your scores.

We are happy to engage in more discussions if you have any further questions/concerns about our paper and our replies.

Best,

Authors

---

### Decision · Program_Chairs · 2022-01-20

**Decision:**

Accept (Poster)

**Comment:**

This paper proposes augmenting standard forward prediction techniques used for representation learning with backward prediction as well, termed "learning via retracing". The paper implements this idea in a Cycle-Consistency World Model (CCWM) and demonstrates that CCWM improves performance of a Dreamer agent across a number of Control Suite tasks. The paper also proposes a way to detect "irreversible" transitions and exclude them from the backwards prediction step.

This paper generated mixed opinions, and the reviewers did not come to a consensus on whether it should be accepted or rejected. In particular, Reviewer VSAG maintained it should be accepted, while Reviewer NEVM maintained it should be rejected. The other reviewers did not reply; I thought the authors' responses to their questions were reasonable so I assume their concerns were addressed (additionally, I don't believe a comparison to PlayVirtual is a justifiable request per the [reviewing guidelines](https://iclr.cc/Conferences/2022/ReviewerGuide), as it is concurrent work).

The reviewers generally agreed that the cycle-consistency idea proposed by the paper is interesting, well-motivated, and borne out by the experimental results. I agree with these points. The main weakness of the paper, brought up by multiple reviewers, was the justification/motivation for the method for irreversibility detection. The authors clarified in the rebuttal that the motivation comes from the idea that temporally-adjacent states with very different values will tend to be far apart in representation space. While I believe that is true, it's not at all clear to me that this necessarily *entails* irreversible transitions between those states. That said, my feeling is that this is approach is (1) not the main contribution of the paper and (2) empirically seems to work, based on the experiments, even if the motivation is unclear/unjustified. Therefore, I do not think the concern about irreversibility detection is grounds on its own for rejection.

Overall, I find this is a sensible approach to better representation learning in MBRL. I recommend acceptance as a poster.